# STRUCTURAL FAIRNESS-AWARE ACTIVE LEARNING FOR GRAPH NEURAL NETWORKS

**Haoyu Han**[1], **Xiaorui Liu**[2], **Li Ma**[3], **MohamadAli Torkamani**[4][*] **Hui Liu**[1],
**Jiliang Tang**[1], **Makoto Yamada**[5]
[1]Michigan State University    [2]North Carolina State University    [3]Shanghai Jiaotong University
[4]Amazon    [5]Okinawa Institute of Science and Technology
{hanhaoy1, liuhui7, tangjili}@msu.edu, xliu96@ncsu.edu, mali-cs@sjtu.edu.cn
alitor@amazon.com, makoto.yamada@oist.jp

## ABSTRACT

Graph Neural Networks (GNNs) have seen significant achievements in semi-supervised node classification. Yet, their efficacy often hinges on access to high-quality labeled node samples, which may not always be available in real-world scenarios. While active learning is commonly employed across various domains to pinpoint and label high-quality samples based on data features, graph data present unique challenges due to their intrinsic structures that render nodes non-i.i.d. Furthermore, biases emerge from the positioning of labeled nodes; for instance, nodes closer to the labeled counterparts often yield better performance. To better leverage graph structure and mitigate structural bias in active learning, we present a unified optimization framework (SCARCE), which is also easily incorporated with node features. Extensive experiments demonstrate that the proposed method not only improves the GNNs performance but also paves the way for more fair results.

## 1 INTRODUCTION

Graphs, representing relationships between entities, are foundational data structures that are wildly used across fields, from recommendation systems (Wu et al., 2022) and fraud detection (Dou et al., 2020) to biology (Molho et al., 2022). Graph Neural Networks (GNNs) have demonstrated impressive capabilities (Gilmer et al., 2017; Kipf & Welling, 2016; Veličković et al., 2017) across various graph tasks, particularly in semi-supervised node classification. However, their performance largely depends on the presence of high-quality labeled node samples. Unfortunately, obtaining these samples can be both time-consuming and costly in real-world scenarios. Moreover, even with an equal count of labeled nodes, varied training sets can result in large performance disparities in GNNs (Fey & Lenssen, 2019; Madhawa & Murata, 2020).

Active Learning (AL) (Ren et al., 2021; Settles, 2011) tackles this issue by identifying the most representative samples for annotation, aiming to maximize model performance within a given labeling budget. Although AL has seen significant success with i.i.d data like images and text, adapting it to GNNs introduces unique challenges. First, the graph structure brings the interdependencies among nodes, making individual sample selection more intricate. Second, typical AL methods often involve iteratively selecting an individual or a batch of samples and then retraining the model. This iterative approach becomes impractical for GNNs, especially for larger graphs. The reason is that training GNNs doesn't just involve the training nodes alone; and it also requires consideration of its multi-hop neighbors, adding to the computational demands.

Several AL methods tailored for GNNs have emerged. For instance, AGE (Cai et al., 2017) addresses the first challenge by incorporating uncertainty and representativeness strategies, commonly found in traditional AL, with graph centrality (Müller et al., 1995) for node selection. This approach necessitates multiple model re-training phases, presenting potential scalability concerns. In contrast, FeatProp (Wu et al., 2019) embarks on a distinct path. It first propagates original features through the graph and then employs the K-Means clustering algorithm on the propagated node features. Nodes closest to cluster centers are then selected. As a one-step AL method, FeatProp only needs one node

---

[*]This work does not relate to the author's position at Amazon

selection round prior to GNN training, making it notably efficient. GraphPart (Ma et al., 2022) further enhances FeatProp by incorporating it with graph partitioning, subsequently applying FeatProp to each partition. While these methods can leverage the graph structure and avoid retraining GNNs, they heavily rely on the quality of initial node features. However, these features may be unavailable, noisy, or uninformative. As demonstrated in Section 2, FeatProp even underperforms random selection on some datasets.

Graph structure, on the other hand, inherently captures the relationships between individual nodes, setting GNNs apart from many other models. This naturally prompts the question: Can we exploit the graph structure to select representative nodes for GNNs? Current research offers promising insights. For instance, Cai et al. (2017) underscores the potential of graph centrality metrics like Degree and PageRank (Page et al., 1998) in aiding node selection. In addition, one recent research (Han et al., 2023) uncovers a label position bias in GNNs. The label position bias indicates that unlabeled nodes in close proximity to labeled ones tend to exhibit superior performance using GNNs, which would cause the fairness issue for the nodes far away from labeled nodes. It differs from the traditional bias caused by sensitive features, such as gender and race. The label position bias is unique for graphs and related to the labeling of graphs. Therefore, strategically selecting nodes for annotation based on their position in graphs can mitigate this fairness issue. In Section 2.2, we find that the fair labeling is also related to the model performance. As a result, we can leverage structural-based active learning to boost both performance and fairness.

In this paper, we first study the relation between different criteria with the performance of GNNs. To combine these criteria and amplify the GNNs' efficacy, we introduce a unified optimization framework tailored for active learning within GNNs. By leveraging the unified framework, we design a Structural fairness-aware active learning method (SCARCE) that can concurrently address performance and fairness issues. Additionally, the inherent flexibility of our optimization framework facilitates the integration of node features, which in some contexts, can further enhance the performance. Extensive experiments show that our proposed method not only achieves comparable or even superior performance but also significantly enhances fairness within GNNs.

## 2 PRELIMINARY

In this section, we begin by examining whether the original node features are always beneficial for AL. Surprisingly, we find they can even degrade the quality of node selection in certain datasets. This observation leads us to consider graph structure-based criteria for AL. Subsequently, we explore the label position bias, which not only affects fairness but also can reflect the overall performance of GNNs. Moreover, we introduce a novel metric grounded in the graph structure, termed "Structure Inertia Score", which gauges the representativeness of node selections. Before diving further, we first establish the notations used in this paper and define active learning under the context of GNNs.

**Notations.** We use bold upper-case letters such as $\mathbf{X}$ to denote matrices. $\mathbf{X}_i$ denotes its $i$-th row and $\mathbf{X}_{ij}$ indicates the $i$-th row and $j$-th column element. We use bold lower-case letters such as $\mathbf{x}$ to denote column vectors. Let $\mathcal{G} = (\mathcal{V}, \mathcal{E})$ be a graph, where $\mathcal{V}$ is the node set with $n$ nodes and $\mathcal{E}$ is the edge set. The graph can be represented by an adjacency matrix $\mathbf{A} \in \mathbb{R}^{n \times n}$, where $\mathbf{A}_{ij} > 0$ indicates that there exists an edge between nodes $v_i$ and $v_j$ in $\mathcal{G}$, or otherwise $\mathbf{A}_{ij} = 0$. Let $\mathbf{D} = diag(d_1, d_2, \ldots, d_n)$ be the degree matrix, where $d_i = \sum_j \mathbf{A}_{ij}$ is the degree of node $v_i$. We define the normalized adjacency matrix as $\tilde{\mathbf{A}} = \mathbf{D}^{-\frac{1}{2}} \mathbf{A} \mathbf{D}^{-\frac{1}{2}}$. Furthermore, suppose that each node is associated with a $d$-dimensional feature $\mathbf{x}$ and we use $\mathbf{X} = [\mathbf{x}_1, \ldots, \mathbf{x}_n]^\top \in \mathbb{R}^{n \times d}$ to denote the feature matrix. We use a binary vector $\mathbf{t} \in \mathbb{R}^{n \times 1}$ as the labeling indicator vector, where $\mathbf{t}_i = 1$ if node $i$ is a labeled node, or otherwise $\mathbf{t}_i = 0$.

**Active Learning for GNNs.** Consider a graph $\mathcal{G} = (\mathbf{A}, \mathbf{X})$ accompanied by a small labeled node set, denoted $\mathbf{s}_0$. Given an annotation budget of $b$, the objective of the active learning is to strategically select node subsets $\mathbf{s}_t$ across multiple iterations such that $|\mathbf{s}_0 \cup \mathbf{s}_1 \cup \mathbf{s}_2 \cup \cdots \cup \mathbf{s}_T| = b$, where $T$ represents the total number of iterations. During each iteration $t$, a GNN model $\hat{g}_t$ is usually trained using the accumulated labeled set $\mathbf{s} = \mathbf{s}_0 \cup \mathbf{s}_1 \cup \mathbf{s}_2 \cup \cdots \cup \mathbf{s}_t$ to select the $\mathbf{s}_{t+1}$. The ultimate goal of active learning for GNNs is to minimize the expected classification error of $\hat{g}_T$ on the unlabeled nodes. In this work, we focus on the one-step active learning setup for GNNs (Contardo et al., 2017; Wu et al., 2019; Ma et al., 2022), where $|s_0| = 0$ and $T = 1$. As a result, our method is model-agnostic.

## 2.1 EXAMINATION ON ORIGINAL NODE FEATURES

Both FeatProp (Wu et al., 2019) and GraphPart (Ma et al., 2022) are based on the propagated original node features, such as $\tilde{\mathbf{A}}^2\mathbf{X}$. These methods then employ clustering techniques, selecting the nodes nearest to cluster centers as training nodes. Central to these approaches is an underlying belief that the propagated original feature can align with the representations learned by GNNs. Although notable results were observed on several citation datasets, this assumption might not hold on other types of datasets. To verify this, we compare the FeatProp with a simple random selection strategy on two wildly used Amazon co-purchase graphs (Shchur et al., 2018), i.e., Computers and Photo. For a comprehensive evaluation, we choose two GNN models: GCN (Kipf & Welling, 2016) and APPNP (Klicpera et al., 2018), which cover both coupled and decoupled GNN structure. The overall budget is selected from 20 to 200, and the result of the Computer dataset is shown in Figure 1. From the result, we can find that the performance of FeatProp is similar to or even worse than the random selection. A similar finding can also be found on the Photo dataset in Appendix A.1. As a result, only leveraging the original feature may not be helpful or even degrade the performance of AL. Therefore, we mainly explore the graph structure in this work and then extend our framework with node features.

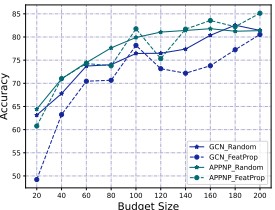

Figure 1: Random vs. FeatProp on Computers.

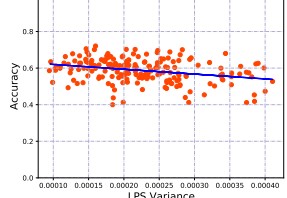

Figure 2: GCN Performance vs. LPS variance on Cora.

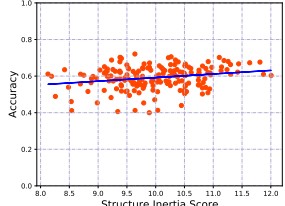

Figure 3: GCN Performance vs. SIS on Cora.

## 2.2 LABEL POSITION BIAS

Label Position Bias (Han et al., 2023) arises when the performance of a GNN model varies for nodes based on their proximity to labeled nodes in the graph, which would cause the performance fairness issue. A key metric used to measure this proximity is the Label Proximal Score defined as follows:

$$LPS = \mathbf{Pt}, \quad \text{and} \quad \mathbf{P} = \left(\mathbf{I} - (1-\alpha)\tilde{\mathbf{A}}\right)^{-1}, \tag{1}$$

where $\mathbf{P} \in \mathbb{R}^{n \times n}$ is the Personalized PageRank matrix, $\mathbf{t} \in \mathrm{R}^{n \times 1}$ is a given labeling indicator vector, $\mathbf{t}_i = 1$ if the node $i$ belong to the training set and otherwise 0, and $\alpha \in (0, 1]$ presents the teleport probability. As observed by Han et al. (2023), unlabeled nodes that possess a higher LPS tend to achieve superior performance in GNNs. They found a low variance of the LPS score, which means all the nodes have a similar "distance" to the labeled node, can mitigate the label position bias and improve the performance fairness of GNNs. Thus, we use the LPS variance to measure structural fairness. The study addressed this by modifying the graph structure $\mathbf{P}$, in a way that reduced the variance in LPS across nodes. This adjustment paved the way for more consistent and fair predictions across different groups of unlabeled nodes with diverse LPS ranges.

In active learning, strategically choosing labeling nodes, represented by $\mathbf{t}$, can potentially reduce the LPS variance, promoting fairness in GNNs. But does this strategic node selection also elevate the overall performance of the GNNs? To validate this, we conducted experiments with the Cora and CiteSeer (Sen et al., 2008) datasets using both GCN and APPNP models. For each experiment, we randomly selected 20 nodes for training and ran the GNNs through three times for every selection. Figure 2 depicts the relationship between LPS variance and GCN accuracy based on 200 random training selections for the Cora dataset. The results of other methods and datasets are detailed in Appendix A.2. Based on these results, we have the following findings: (1) Node selection can profoundly impact GNN performance. For instance, in Figure 2, we observe an accuracy disparity reaching up to 35%.; (2) Selections with better structural fairness(lower LPS variance) typically correspond to superior GNN results. Therefore, beyond just fostering fairness, reduced LPS variance can serve as a crucial criterion for labeling node selection, subsequently enhancing GNN performance.

## 2.3 STRUCTURE INERTIA SCORE

While ensuring a smaller LPS variance can promote fairness and potentially elevate GNN performance, it's important to also emphasize the selection of representative nodes within the graph structure.

Traditionally, graph centrality metrics such as Degree and PageRank have been leveraged in AL to identify these representative nodes. However, they select nodes independently. Consequently, if two adjacent nodes both have high Degree or PageRank scores, they are likely to be selected together, potentially overlooking the broader scope of the graph structure. In recently years, the cluster-based methods (Zhan et al., 2022; Sener & Savarese, 2017; Wu et al., 2019; Ma et al., 2022), have achieved significant success in AL. The foundation of their efficacy is rooted in the selection of samples that reside close to the cluster centers. These centrally located samples can subsequently serve as representative proxies for other samples within their respective clusters. Inspired by this premise, our objective is to select the most representative nodes within graph structure for GNNs. Inertia, a metric that calculates the sum of distances between individual data points and their corresponding centroids, is frequently employed to measure how well the data is clustered. In graph-based AL, we consider labeled nodes as cluster centers and introduce the Structure Inertia Score (SIS), formulated as:

$$SIS = \sum_{i \in \mathcal{V}} \max_{j} \mathbf{P}_{ij} \mathbf{t}_j, \tag{2}$$

where $\mathbf{P}$ represents the Personalized PageRank matrix. Each entry, $\mathbf{P}_{ij} > 0$, captures the structural similarity between node $i$ and node $j$, and $\mathbf{t}$ indicates the node selections. Thus, the Structure Inertia Score encapsulates the cumulative similarity of each node to its proximate labeled node, providing a metric to estimate the representativeness of labeled nodes.

Following the same setting with section 2.2, we explore the relationship between the SIS and the GNNs performance. Figure 3 illustrates the correlation between SIS and GCN accuracy on the Cora dataset, and results for alternative methods and datasets are in Appendix A.3. From these results, the labeling selection with a higher SIS tends to have a higher performance. Therefore, the SIS can also serve as a structural criterion for labeling node selection.

## 3 THE PROPOSED METHOD

From the literature and the preliminary studies in Section 2, there are different criteria for selecting labeling nodes. These criteria might not always be overlapped, and using them together could potentially lead to enhanced performance. However, there are several challenges: (1) How to combine these criteria to boost the final performance of GNNs? (2) There is usually no validation set in AL, and how can we balance each criterion? To address these issues, we propose a unified framework where active learning for GNNs is conceptualized as an optimization problem. This design inherently supports the flexible incorporation of diverse criteria. Additionally, we also introduce a gradient balance method to harmonize the different terms. One significant advantage of our framework is that it obviates the need for hyperparameter tuning. This is especially beneficial for active learning: hyperparameter tuning might inadvertently lead to the annotation of suboptimal nodes, thereby leading to an increased and potentially unnecessary annotation burden. Leveraging our unified optimization framework, we introduce a structural fairness-aware active learning method, namely SCARCE, tailored for GNNs. Additionally, due to the flexibility of the unified framework, it is easy to incorporate node features.

### 3.1 A UNIFIED OPTIMIZATION FRAMEWORK

To conduct the active learning for GNNs, we seek a binary vector $\mathbf{t} \in \mathbb{R}^{n \times 1}$ to indicate whether a node is selected for annotation or not. Thus, the active learning for GNNs can be represented as:

$$\begin{aligned} \underset{\mathbf{t}}{\text{minimize}} \quad & f(\mathcal{G}, \mathbf{t}) \\ \text{s.t.} \quad & \mathbf{t}\mathbf{1}_n^\top \leq b, \mathbf{t} \in \{0,1\}^n, \end{aligned} \tag{3}$$

where the $f = \mathcal{L}_1 + \mathcal{L}_2 + \cdots + \mathcal{L}_l$ represents the loss function of all $l$ different criteria, and $\mathbf{1}_n \in \mathbb{R}^{n \times 1}$ is all one vector. Following Xu et al. (2019), we can ease this combinatorial optimization problem by relaxing the binary vector $\mathbf{t}$ to its convex hull $\mathcal{S} = \{\mathbf{t}|\mathbf{t} \in [0,1]^n, \mathbf{t}\mathbf{1}_n^\top \leq b\}$, which can be solved by the projected gradient descent (PGD) method (Boyd & Vandenberghe, 2004):

$$\mathbf{t}^{(k)} = \Pi_{\mathcal{S}} \left[ \mathbf{t}^{(k-1)} - \eta_k \nabla_{\mathbf{t}} f \left( \mathcal{G}, \mathbf{t}^{(k-1)} \right) \right], \tag{4}$$

where $k$ is the current iteration step, $\eta_k$ represents the learning rate at the step $k$, and $\nabla_{\mathbf{t}} f \left( \mathcal{G}, \mathbf{t}^{(k-1)} \right)$ denotes the gradient of the loss function $f$ at $\mathbf{t}^{(k-1)}$. Besides, $\Pi_{\mathcal{S}}[\mathbf{t}] = \arg\min_{\mathbf{s} \in \mathcal{S}} \|\mathbf{s} - \mathbf{t}\|_2^2$ is a projection operator that projects $\mathbf{t}$ into the constraint set $\mathcal{S}$.

**Gradient Balance.** One essential issue of this optimization process is how to balance different criteria. Due to our focus on the one-step active learning for GNNs, there are no available labeled nodes as the validation set. This absence complicates the fine-tuning of hyperparameters to balance various terms $\mathcal{L}_1, \mathcal{L}_2, \cdots, \mathcal{L}_l$ effectively. Furthermore, each round of hyperparameter tuning may necessitate the annotation of non-ideal nodes. If we use the same weight for different criteria, the gradient scale of each criterion might exhibit different scales. As as result, the optimization trajectory may be overwhelmingly influenced by the term with the larger gradient scale, potentially leading to imbalanced learning. Inspired by the multi-task learning (Liu et al., 2021; Guo & Wei, 2022), where adaptive gradient mechanisms are employed to balance the contribution of multiple tasks, we introduce a simple but efficient gradient balance method to balance different terms. Specifically, The process is twofold: Firstly, we normalize the gradient scale of each term, resulting in $\mathbf{u}_i = \frac{\nabla_{\mathbf{t}} \mathcal{L}_i}{\|\nabla_{\mathbf{t}} \mathcal{L}_i\|_1}$ to ensure a unit scale. Subsequently, we uniformly scale every gradient using the factor $s = \min_i \|\nabla_{\mathbf{t}} \mathcal{L}_i\|_1$. As a result, the gradients of different terms should be in the same order of magnitude.

After obtaining the optimal continuous vector $\mathbf{t}^*$, the $i$-th element $\mathbf{t}_i^*$ represents the probability that the node $i$ is selected as the labeling node. Finally, we can generate the binary labeling indicator vector $\mathbf{t}$ by sampling from $\mathbf{t}^*$.

## 3.2 STRUCTURAL FAIRNESS-AWARE AL FOR GNNS

The studies in Section 2 provide some insights into active learning for GNNs. First, the original node features may not align well with the representations learned by GNNs in some datasets, even after several iterations of feature propagation over graphs. This discrepancy underscores the importance of leveraging the graph structure to select the most representative nodes for GNNs. Second, the label proximity score not only reflects the disparity of GNNs performance in different groups of nodes, where nodes with higher LPS tend to perform better, but its variance can also indicate the performance of GNNs. Specifically, node selections with low LPS variance tend to yield superior GNN performance. Third, drawing inspiration from clustering algorithms, the Structure Inertia Score can serve as an indicator of the representational potency of labeled nodes within the graph structure. Node selections that boast a higher SIS generally pave the way for more effective GNN training. Consequently, there's potential to enhance GNN performance by identifying nodes that exhibit both a low variance in LPS and a high SIS.

The unified optimization framework enables us to incorporate both LPS variance and SIS to improve the fairness and performance of GNNs. Therefore, the corresponding objective function can be represented by:

$$\arg\min_{\mathbf{t}} f(\mathcal{G}, \mathbf{t}) = \|\mathbf{P}\mathbf{t} - c\mathbf{1}_n\|_2^2 - \sum_{i \in \mathcal{V}} \max_j \mathbf{P}_{ij}\mathbf{t}_j, \tag{5}$$

where $c = \frac{(\mathbf{P}\mathbf{t})^\top \mathbf{1}_n}{n}$ represents the mean of label position score $\mathbf{P}\mathbf{t}$. The gradient of the first term respect to $\mathbf{t}$ can be represented by:

$$\nabla_{\mathbf{t}} \mathcal{L}_1 = \nabla_{\mathbf{t}} \left\| \mathbf{P}\mathbf{t} - \frac{(\mathbf{P}\mathbf{t})^\top \mathbf{1}_n}{n} \mathbf{1}_n \right\|_2^2 = 2\mathbf{P}^\top \mathbf{P}\mathbf{t} - 2\frac{(\mathbf{P}\mathbf{t})^\top \mathbf{1}_n}{n} \mathbf{P}^\top \mathbf{1}_n, \tag{6}$$

and the gradient of the second term with respect to $\mathbf{t}_k$ is:

$$\nabla_{\mathbf{t}_k} \mathcal{L}_2 = -\nabla_{\mathbf{t}_k} \sum_{i \in \mathcal{V}} \max_j \mathbf{P}_{ij}\mathbf{t}_j = -\sum_{i \in \mathcal{V}} \begin{cases} P_{ik}, & \text{if } j = k \text{ for that } i, \\ 0, & \text{otherwise.} \end{cases} \tag{7}$$

Finally, we can update the $\mathbf{t}$ using the PGD in Eq. (4) with the gradient balance method. Building upon the fact that this method solely relies on the graph structure, we name it as **SCARCE** or **SCARCE-Structure**. The full algorithm is detailed in Algorithm 1. First, we uniformly initialize $\mathbf{t}$ in line 3. Next, we perform projected gradient descent with gradient balance from lines 4-9. We then sample potential labeling selections using a multinomial sampling method. Finally, in line 14, we select the binary vector $\mathbf{t}$ that yields the smallest overall loss.

## 3.3 SCARCE WITH NODE FEATURES

Even though the original node features might not always be informative, studies by Wu et al. (2019); Ma et al. (2022) have demonstrated their potential utility in certain datasets. The flexibility of our

---

**Algorithm 1** Algorithm of SCARCE-Structure.

---

1: **Input:** The PPR matrix $\mathbf{P}$, budget b, learning rate $\eta$, the iteration numbers $K$, and the number of random trials $R$
2: **Output**: A binary labeling indicator vector $\mathbf{t}$ with $|\mathbf{t}| = b$
3: **Initialization**: $\mathbf{t}^{(0)} = [b/n, b/n, \ldots, b/n]^\top$
4: **for** $k = 1, 2, \ldots, K$ **do**
5:     calculate $\nabla_{\mathbf{t}^{(k-1)}} \mathcal{L}_1$ and $\nabla_{\mathbf{t}^{(k-1)}} \mathcal{L}_2$ based on Eq. (6) and Eq. (7)
6:     normalize the gradient $\mathbf{u}_1^{(k-1)} = \frac{\nabla_{\mathbf{t}^{(k-1)}} \mathcal{L}_1}{\|\nabla_{\mathbf{t}^{(k-1)}} \mathcal{L}_1\|_1}$ and $\mathbf{u}_2^{(k-1)} = \frac{\nabla_{\mathbf{t}^{(k-1)}} \mathcal{L}_2}{\|\nabla_{\mathbf{t}^{(k-1)}} \mathcal{L}_2\|_1}$
7:     $s = \min\left(\|\nabla_{\mathbf{t}^{(k-1)}} \mathcal{L}_1\|_1, \|\nabla_{\mathbf{t}^{(k-1)}} \mathcal{L}_2\|_1\right)$
8:     projected gradient descent: $\mathbf{t}^{(k)} = \Pi_{\mathcal{S}}\left[\mathbf{t}^{(k-1)} - \eta s \left(\mathbf{u}_1^{k-1} + \mathbf{u}_2^{k-1}\right)\right],$
9: **end for**
10: normalize $\hat{\mathbf{t}} = \frac{\mathbf{t}^{(K)}}{\|\mathbf{t}^{(K)}\|_1}$
11: **for** $r = 1, 2, 3, \ldots, R$ **do**
12:     draw binary vector $\mathbf{s}^{(r)}$ by multinomial sampling: $\mathbf{s}^{(r)} = \text{multinoimal}(\hat{\mathbf{t}}, b)$
13: **end for**
14: choose a binary vector $\mathbf{t}$ from $\{\mathbf{s}^{(r)}\}$ which yields the smallest loss in Eq. (5)
15: **return** $\mathbf{t}$

---

unified optimization framework facilitates the seamless integration of these features into SCARCE. Specifically, similar to the Structure Inertia Score, we design a Feature Inertia Score (FIS):

$$FIS = \sum_{i \in \mathcal{V}} \max_j \mathbf{S}_{ij} \mathbf{t}_j, \quad \mathbf{S}_{ij} = \frac{\exp\left(-\|g(\mathbf{X}_i) - g(\mathbf{X}_j)\|_2^2\right)}{\sum_{l=1}^n \exp\left(-\|g(\mathbf{X}_i) - g(\mathbf{X}_l)\|_2^2\right)}, \tag{8}$$

where the $\mathbf{S}_{i,j}$ represents the feature similarity between nodes $i$ and $j$, and $g(\mathbf{X})$ is a feature propagation function. We have the option to exclusively utilize FIS for labeling node selection, namely as **SCARCE-Feature**, or to integrate it with the structure, referred to as **SCARCE-ALL**.

### 3.4 COMPLEXITY ANALYSIS

The inputs in Algorithm 1 include a Personalized PageRank matrix $\mathbf{P} \in \mathbb{R}^{n \times n}$, which can potentially be a dense matrix. To address this, we adopt the approach of PPRGo (Bojchevski et al., 2020) to calculate a sparse approximate PPR matrix $\mathbf{P}^{(\epsilon)}$, where $\epsilon$ is the threshold, and each row of $\mathbf{P}^{(\epsilon)}$ only contains the top k largest entries. Therefore, the time complexity to calculate the gradient in Eq. (6) and Eq. (7) becomes $O(nk)$, which is linear with respect to the number of nodes. Similarly, when incorporating node features, we employ a strategy that preserves only the top k largest entries in each row of $\mathbf{S}$. Overall, the proposed method is applicable to large graphs.

## 4 EXPERIMENT

### 4.1 EXPERIMENTAL SETTINGS

In our experiments, we evaluate the performance of various AL methods across multiple datasets and investigate their adaptability with different backbone GNNs for the semi-supervised node classification task. Detailed settings are elaborated upon below.

**Datasets & Backbone GNNs.** We perform experiments utilizing six widely used real-world graph datasets, encompassing three citation datasets, i.e., Cora, Citeseer, and Pubmed (Sen et al., 2008), two co-purchase datasets from Amazon, i.e., Computers and Photo (Shchur et al., 2018), and one OGB dataset, i.e., ogbn-arxiv (Hu et al., 2020). The summary details pertaining to these datasets are provided in Appendix B.1. We also adopt different GNN backbones, i.e., GCN (Kipf & Welling, 2016) and APPNP (Klicpera et al., 2018), GAT (Velickovic et al., 2017) and GCNII (Chen et al., 2020).

**Baselines.** We compare our methods with the following baselines, which can be divided into three categories: (1) General AL methods, including Random, Uncertainty (Settles & Craven, 2008), Density (Cai et al., 2017), and CoreSet (Sener & Savarese, 2018); (2) Structure only methods, including Degree and PageRank (Cai et al., 2017); (3) AL methods for GNNs that combine both structure and features of graph, including AGE (Cai et al., 2017), FeatProp (Wu et al., 2019) and GraphPart (Ma et al., 2022).

**Active Learning Setup.** For each dataset, we set the annotation budget to $ZC$ for all methods, where $C$ is the number of classes in the dataset, and $Z$ is chose from $\{5, 10, 20\}$. Following the AL setting in Ma et al. (2022), we set the initial seed set $\mathbf{s}_0$ to zero for the one-step AL methods, such as Random, Degree, PageRank, FeatProp, and GraphPart. For the other iterative AL methods, we randomly choose one-third of the budget as the initial seed set $\mathbf{s}_0$, and let the method select the other two-thirds. Due to the lack of validation set in the AL setup, we train the GNN model with fixed 300 epochs and evaluate over the full graph. For each experiment, we train the backbone GNNs 10 times and report the average performance and standard deviation. For the baselines, we follow the setting in Ma et al. (2022). For our SCARCE, we choose the learning rate $\eta$ from $\{0.001, 0.01, 0.1\}$ based on the gradient scale of $\mathbf{t}$. More details about the experimental settings and GNN architectures can be found in Appendix B.2.

## 4.2 Performance Comparison on Benchmark Datasets

In this subsection, we evaluate the efficacy of the three variations of our proposed SCARCE across all six datasets. Specifically, we examine SCARCE-Structure, which relies solely on the graph structure, SCARCE-Feature, which exclusively employs the propagated features, and SCARCE-ALL, which integrates all criteria. The performance results for GCN and APPNP are presented in Table 1 and Table 2, respectively. From the results, we can make the following observations:

- The Active Learning methods designed for graphs, such as AGE, FeatProp, GraphPart, and our SCARCE, usually perform better than the General AL methods. This underscores the importance of the distinctive graph structures.

- The proposed SCARCE-Structure consistently outperforms the graph centrality-based methods, such as Degree and PageRank. Impressively, this structure-based method can have comparable or even better results compared with the state-of-the-art baselines, such as FeatProp and GraphPart, which exploit both structure and feature information. Notably, on the Computer and Photo datasets, SCARCE-Structure significantly outperforms other baselines. For example, when $Z$=5, it achieves an 8.3% and 7.1% relative performance boost over the best baselines on the Computers dataset, when employing the GCN and APPNP models, respectively.

- The SCARCE-Feature can achieve similar performance with FeatProp and GraphPart across all datasets and GNNs. This attests to the effectiveness of our proposed Feature Inertia Score.

- By integrating both node features and graph structure, SCARCE-ALL typically enhances performance compared to solely leveraging feature or structure on the Cora and CiteSeer datasets. This underscores the potential of combining features and structure in graph-based active learning. However, on the Computers and Photo datasets, SCARCE-ALL performance lags behind the structure-only variant. Besides, the performance of SCARCE-Feature is also subpar, similar to other feature-based methods, such as FeatProp and GraphPart. This observation is consistent with our earlier discussions in the preliminary study, which highlighted that the performance of these methods heavily relies on the quality of the original feature.

To further demonstrate the superiority of the proposed SCARCE, we conduct experiments under the noise feature setting and missing feature setting. The results are shown in Appendix B.3.

## 4.3 Fairness Comparison

In this subsection, we aim to assess the capability of the proposed SCARCE in alleviating the label position bias issue, aiming to bolster the fairness of GNNs. We deploy experiments on both the CiteSeer and Computers datasets using GCN and APPNP models. Specifically, following Han et al. (2023), we first sort the nodes by their LPS in Eq. 1. Subsequently, these nodes are divided into ten distinct sensitivity groups, ensuring each group comprises an equal number of nodes. To evaluate the disparity in performance across these groups, we rely on two commonly used metrics for assessing group bias: the Standard Deviation ($SD$) and the Coefficient of Variation ($CV$). Here, $CV$ is defined as the ratio of $SD$ to $\mu$, where $\mu$ represents the average accuracy across all groups. The results of CiteSeer and Computers datasets are shown in Figure 4 and Figure 5, respectively. We can find:

- The proposed SCARCE consistently yields the lowest SD and CV values across most budget sizes on both the CiteSeer and Computers datasets, irrespective of whether GCN or APPNP is used. Consequently, nodes across diverse sensitivity groups benefit from a more balanced and equitable classification performance when processed by GNNs. This highlights the capability of SCARCE to enhance not just the performance, but also the fairness of GNNs.

Table 1: Semi-supervised node classification accuracy with GCN on benchmark datasets. The **bold** marker denotes the best performance and the underlined marker denotes the second-best performance.

| Baselines Budget = $C*Z$ | Cora | | | Citeseer | | | Pubmed | | |
|---|---|---|---|---|---|---|---|---|---|
| | $Z=5$ | $Z=10$ | $Z=20$ | $Z=5$ | $Z=10$ | $Z=20$ | $Z=5$ | $Z=10$ | $Z=20$ |
| Random | 67.08 ± 4.76 | 76.48 ± 3.42 | 81.09 ± 1.60 | 52.76 ± 7.64 | 63.42 ± 1.94 | 68.49 ± 1.39 | 61.43 ± 6.04 | 67.14 ± 4.11 | 77.21 ± 1.95 |
| Uncertainty | 59.54 ± 5.75 | 70.97 ± 3.44 | 78.12 ± 2.05 | 42.02 ± 9.64 | 54.08 ± 4.43 | 63.59 ± 2.20 | 58.13 ± 5.14 | 60.71 ± 8.36 | 72.47 ± 4.05 |
| Density | 65.71 ± 6.14 | 74.65 ± 2.65 | 80.15 ± 1.34 | 52.46 ± 5.13 | 60.47 ± 4.55 | 67.51 ± 1.31 | 58.64 ± 6.92 | 67.92 ± 2.13 | 74.48 ± 3.54 |
| CoreSet | 64.06 ± 4.16 | 74.94 ± 1.95 | 79.73 ± 1.19 | 51.39 ± 3.65 | 61.73 ± 2.95 | 68.03 ± 1.29 | 56.29 ± 6.04 | 70.27 ± 4.89 | 74.03 ± 2.80 |
| Degree | 64.75 ± 0.65 | 75.99 ± 0.70 | 78.49 ± 0.42 | 46.79 ± 0.95 | 47.68 ± 0.45 | 51.73 ± 0.42 | 61.54 ± 0.49 | 64.19 ± 1.14 | 62.70 ± 1.33 |
| Pagerank | 62.42 ± 1.16 | 75.11 ± 0.29 | 82.63 ± 0.19 | 48.98 ± 0.62 | 62.34 ± 0.29 | 69.14 ± 0.19 | 57.83 ± 0.21 | 66.68 ± 0.22 | 74.36 ± 0.14 |
| AGE | 69.19 ± 3.20 | 76.32 ± 1.12 | 82.14 ± 0.91 | 53.42 ± 5.20 | 59.95 ± 5.28 | 68.77 ± 1.28 | 62.08 ± 7.76 | 69.33 ± 3.05 | 77.56 ± 1.59 |
| FeatProp | 74.49 ± 0.48 | 79.23 ± 0.48 | 83.96 ± 0.20 | 47.98 ± 0.60 | 68.44 ± 0.22 | 70.62 ± 0.14 | 66.49 ± 0.29 | 71.47 ± 0.25 | 79.95 ± 0.09 |
| GraphPart | 78.59 ± 0.43 | **81.79 ± 0.24** | 84.29 ± 0.23 | 63.71 ± 0.52 | 65.85 ± 0.22 | 70.91 ± 0.15 | 72.18 ± 0.31 | **79.08 ± 0.14** | 79.42 ± 0.40 |
| SCARCE-Structure | 76.92 ± 0.74 | 81.55 ± 0.56 | 83.32 ± 0.41 | 64.88 ± 1.32 | 70.26 ± 0.15 | 72.58 ± 0.23 | 72.31 ± 0.75 | 75.65 ± 0.76 | 78.08 ± 1.13 |
| SCARCE-Feature | 77.24 ± 0.78 | 81.09 ± 0.35 | 84.04 ± 0.17 | 65.01 ± 1.23 | 69.06 ± 0.52 | 71.82 ± 0.03 | 71.31 ± 0.55 | 78.40 ± 0.69 | **80.34 ± 0.29** |
| SCARCE-ALL | **80.58 ± 0.82** | 81.67 ± 0.37 | **84.84 ± 0.26** | **66.03 ± 0.32** | **71.93 ± 0.40** | **74.10 ± 0.46** | **74.01 ± 0.38** | 77.01 ± 0.19 | 79.15 ± 0.82 |

| Baselines Budget = $C*Z$ | Computers | | | Photo | | | Arxiv | | |
|---|---|---|---|---|---|---|---|---|---|
| | $Z=5$ | $Z=10$ | $Z=20$ | $Z=5$ | $Z=10$ | $Z=20$ | $Z=5$ | $Z=10$ | $Z=20$ |
| Random | 72.94 ± 4.89 | 79.92 ± 2.01 | 84.18 ± 1.19 | 76.62 ± 10.59 | 85.80 ± 3.48 | 88.57 ± 3.52 | | 57.31 ± 0.98 | 60.41 ± 1.02 |
| Uncertainty | 64.79 ± 8.83 | 75.80 ± 4.04 | 81.17 ± 1.47 | 73.14 ± 8.46 | 82.27 ± 5.13 | 85.87 ± 8.13 | 47.83 ± 1.91 | 53.50 ± 0.93 | 58.44 ± 1.22 |
| Density | 64.81 ± 3.55 | 72.86 ± 4.24 | 82.16 ± 1.75 | 73.94 ± 6.14 | 84.04 ± 4.09 | 89.12 ± 1.65 | 47.12 ± 2.06 | 51.92 ± 1.81 | 55.16 ± 1.35 |
| CoreSet | 56.12 ± 14.37 | 62.73 ± 12.41 | 64.08 ± 15.17 | 60.07 ± 20.03 | 71.87 ± 15.35 | 64.59 ± 23.87 | 48.38 ± 1.17 | 53.81 ± 1.22 | 57.66 ± 0.80 |
| Degree | 46.85 ± 4.21 | 50.14 ± 3.14 | 61.52 ± 8.87 | 40.32 ± 4.13 | 52.80 ± 0.67 | 71.74 ± 18.05 | 37.85 ± 1.78 | 41.76 ± 1.18 | 46.39 ± 0.82 |
| Pagerank | 73.42 ± 2.07 | 76.26 ± 8.23 | 84.52 ± 0.78 | 81.07 ± 2.76 | 81.48 ± 10.70 | 90.68 ± 1.14 | 52.83 ± 0.69 | 57.40 ± 0.38 | 61.11 ± 0.36 |
| AGE | 72.80 ± 4.55 | 75.82 ± 9.99 | 83.34 ± 4.14 | 80.84 ± 3.17 | 86.15 ± 4.51 | 89.75 ± 2.83 | 51.11 ± 1.70 | 56.46 ± 0.78 | 60.05 ± 0.99 |
| FeatProp | 64.68 ± 14.19 | 59.54 ± 15.30 | 76.95 ± 11.11 | 74.48 ± 9.23 | 87.24 ± 0.62 | 88.41 ± 4.63 | 54.52 ± 0.64 | 56.84 ± 0.59 | 60.49 ± 0.50 |
| GraphPart | 64.07 ± 4.23 | 72.08 ± 5.92 | 79.67 ± 8.77 | 71.19 ± 10.62 | 82.52 ± 6.47 | 87.13 ± 1.31 | 54.52 ± 1.31 | 55.74 ± 0.55 | 59.74 ± 0.45 |
| SCARCE-Structure | **79.51 ± 1.90** | **83.94 ± 0.44** | **86.05 ± 0.39** | **87.65 ± 2.19** | **90.13 ± 1.01** | **91.37 ± 0.64** | **56.26 ± 0.53** | 58.32 ± 0.35 | 61.05 ± 0.32 |
| SCARCE-Feature | 67.32 ± 1.98 | 73.93 ± 1.16 | 78.81 ± 1.51 | 77.60 ± 2.49 | 85.15 ± 0.93 | 87.83 ± 0.92 | 54.62 ± 0.31 | 58.80 ± 0.36 | **61.98 ± 0.22** |
| SCARCE-ALL | 76.52 ± 1.61 | 81.82 ± 1.02 | 84.85 ± 0.65 | 85.65 ± 1.37 | 89.13 ± 0.75 | 90.95 ± 0.31 | 54.84 ± 0.40 | **58.98 ± 0.27** | 61.54 ± 0.36 |

Table 2: Semi-supervised node classification accuracy with APPNP on benchmark datasets. The **bold** marker denotes the best performance and the underlined marker denotes the second-best performance.

| Baselines Budget = $C*Z$ | Cora | | | CiteSeer | | | PubMed | | |
|---|---|---|---|---|---|---|---|---|---|
| | $Z=5$ | $Z=10$ | $Z=20$ | $Z=5$ | $Z=10$ | $Z=20$ | $Z=5$ | $Z=10$ | $Z=20$ |
| Random | 70.25 ± 5.17 | 78.16 ± 3.75 | 83.05 ± 1.18 | 54.99 ± 8.11 | 65.37 ± 2.38 | 68.73 ± 1.13 | 65.58 ± 6.37 | 70.98 ± 4.15 | 78.73 ± 1.80 |
| Uncertainty | 64.49 ± 5.74 | 75.83 ± 3.26 | 80.44 ± 2.42 | 47.53 ± 7.58 | 55.60 ± 4.47 | 66.48 ± 1.54 | 58.17 ± 5.19 | 63.95 ± 8.93 | 75.44 ± 4.27 |
| Density | 70.69 ± 4.52 | 77.70 ± 2.78 | 82.31 ± 1.25 | 54.53 ± 6.94 | 61.94 ± 3.63 | 67.05 ± 1.52 | 62.15 ± 6.92 | 70.10 ± 6.90 | 77.58 ± 2.66 |
| CoreSet | 69.12 ± 2.68 | 76.37 ± 2.93 | 82.31 ± 0.88 | 54.43 ± 6.35 | 64.19 ± 4.09 | 68.69 ± 1.46 | 66.72 ± 6.54 | 71.75 ± 4.42 | 77.78 ± 2.41 |
| Degree | 68.52 ± 0.40 | 77.09 ± 0.24 | 80.26 ± 0.24 | 53.47 ± 0.75 | 52.41 ± 0.92 | 56.80 ± 0.39 | 62.12 ± 0.40 | 66.16 ± 0.15 | 65.28 ± 0.15 |
| Pagerank | 70.39 ± 0.70 | 76.92 ± 0.16 | 83.90 ± 0.18 | 51.51 ± 1.00 | 63.17 ± 0.26 | 70.21 ± 0.20 | 60.22 ± 0.18 | 70.49 ± 0.23 | 77.19 ± 0.08 |
| AGE | 72.05 ± 3.10 | 78.57 ± 1.81 | 83.43 ± 0.72 | 52.95 ± 5.45 | 62.59 ± 4.29 | 69.83 ± 1.62 | 59.83 ± 7.94 | 69.40 ± 5.05 | 79.05 ± 1.46 |
| FeatProp | 77.81 ± 0.34 | 81.47 ± 0.36 | 84.43 ± 0.12 | 48.92 ± 0.37 | 67.98 ± 0.18 | 72.47 ± 0.13 | 69.83 ± 0.16 | 75.16 ± 0.04 | 80.40 ± 0.10 |
| GraphPart | 80.38 ± 0.40 | 83.22 ± 0.15 | 84.79 ± 0.16 | 64.83 ± 0.38 | 66.89 ± 0.27 | 70.33 ± 0.20 | 74.56 ± 0.04 | **80.34 ± 0.03** | 82.39 ± 0.06 |
| SCARCE-Structure | 79.47 ± 0.83 | **83.35 ± 0.32** | 84.45 ± 0.36 | 65.94 ± 1.00 | 66.09 ± 1.19 | 70.83 ± 0.62 | 74.12 ± 1.64 | 80.01 ± 0.81 | 81.40 ± 0.43 |
| SCARCE-Feature | 77.92 ± 1.06 | 82.54 ± 0.37 | 83.91 ± 0.19 | 65.28 ± 0.73 | 66.75 ± 1.27 | 71.24 ± 0.41 | 72.95 ± 1.55 | 73.12 ± 0.87 | 79.29 ± 0.63 |
| SCARCE-ALL | **82.77 ± 0.69** | 83.17 ± 0.30 | **84.99 ± 0.43** | **67.15 ± 0.58** | **68.43 ± 0.94** | **72.50 ± 0.36** | **76.01 ± 0.89** | 78.85 ± 0.95 | 80.66 ± 0.37 |

| Baselines Budget = $C*Z$ | Computers | | | Photo | | | Arxiv | | |
|---|---|---|---|---|---|---|---|---|---|
| | $Z=5$ | $Z=10$ | $Z=20$ | $Z=5$ | $Z=10$ | $Z=20$ | $Z=5$ | $Z=10$ | $Z=20$ |
| Random | 74.06 ± 4.87 | 76.30 ± 10.05 | 83.46 ± 1.42 | 81.90 ± 6.03 | 88.47 ± 1.50 | 90.36 ± 3.90 | 54.85 ± 0.75 | 58.41 ± 1.19 | 61.91 ± 0.68, |
| Uncertainty | 65.52 ± 4.89 | 71.92 ± 4.63 | 79.00 ± 4.88 | 78.88 ± 4.79 | 81.20 ± 5.84 | 89.20 ± 3.68 | 49.16 ± 2.12 | 54.42 ± 1.61 | 58.75 ± 0.40, |
| Density | 67.72 ± 6.35 | 74.63 ± 3.17 | 77.39 ± 10.67 | 75.31 ± 8.01 | 87.36 ± 2.67 | 90.23 ± 1.14 | 48.84 ± 1.68 | 54.07 ± 1.72 | 56.22 ± 1.22, |
| CoreSet | 65.38 ± 6.05 | 71.47 ± 6.29 | 76.12 ± 5.45 | 74.55 ± 11.11 | 87.82 ± 1.38 | 89.08 ± 3.58 | 50.31 ± 1.49 | 55.94 ± 0.55 | 58.79 ± 1.67, |
| Degree | 44.89 ± 2.72 | 51.22 ± 2.84 | 64.42 ± 4.64 | 42.08 ± 3.67 | 57.78 ± 2.55 | 77.67 ± 12.99 | 40.36 ± 1.49 | 43.81 ± 0.90 | 48.40 ± 0.88, |
| Pagerank | 74.86 ± 9.59 | 79.89 ± 3.16 | 81.56 ± 7.31 | **85.85 ± 1.04** | 87.09 ± 0.24 | 90.49 ± 1.16 | 54.75 ± 0.44 | 59.42 ± 0.30 | 62.33 ± 0.30, |
| AGE | 71.67 ± 4.47 | 76.91 ± 10.89 | 84.85 ± 0.66 | 81.47 ± 4.95 | 87.33 ± 1.73 | 90.94 ± 1.03 | 52.36 ± 1.86 | 56.46 ± 1.23 | 59.43 ± 1.06, |
| FeatProp | 74.55 ± 5.41 | 79.16 ± 2.41 | 80.05 ± 6.22 | 83.75 ± 1.31 | 88.33 ± 0.63 | 91.04 ± 0.62 | 55.74 ± 0.71 | 58.17 ± 0.53 | 61.96 ± 0.62 |
| GraphPart | 67.81 ± 1.29 | 72.60 ± 2.66 | 80.76 ± 4.32 | 78.84 ± 3.22 | 88.62 ± 1.03 | 89.81 ± 0.74 | 55.09 ± 0.67 | 56.98 ± 0.69 | 61.05 ± 0.73 |
| SCARCE-Structure | **80.18 ± 0.99** | **84.44 ± 0.95** | **85.85 ± 0.47** | 85.78 ± 2.52 | 88.76 ± 1.03 | 91.50 ± 1.01 | **57.10 ± 0.29** | 58.95 ± 0.34 | **62.88 ± 0.12** |
| SCARCE-Feature | 65.37 ± 3.30 | 68.29 ± 1.79 | 77.60 ± 2.29 | 80.73 ± 0.85 | 87.58 ± 1.73 | 90.34 ± 1.27 | 56.01 ± 0.40 | 60.35 ± 0.26 | 62.58 ± 0.15 |
| SCARCE-ALL | 77.10 ± 1.04 | 80.33 ± 1.09 | 85.59 ± 1.05 | 85.75 ± 2.65 | **89.77 ± 1.55** | **91.53 ± 1.32** | 56.19 ± 0.24 | **60.96 ± 0.33** | 62.24 ± 0.31 |

- The degree-based method usually exhibits high SD and CV values. This can be attributed to the fact that neighbors of these labeled high-degree nodes often possess a higher LPS. Consequently, these nodes tend to outperform others, leading to discrepancies in classification performance across the graph.

## 4.4 ABLATION STUDY

In this subsection, we investigate the impact of LPS variance and SIS within the SCARCE on GNN performance and fairness. Specifically, we introduce two variants that exclusively incorporate either the LPS variance or SIS into the unified framework. Subsequently, we assess their implications on both accuracy and fairness using the GCN model. Results for the Cora and CiteSeer datasets across varying budgets are depicted in Fig.6 and Fig.7, respectively. From the results, several observations can be made: (1) Leveraging SIS independently tends to surpass the performance achieved by solely using LPS variance, particularly on the CiteSeer dataset. (2) However, LPS variance usually leads to a lower SD, indicating its inherent capacity to promote increased fairness. (3) SCARCE, which combines both SIS and LPS variance, SCARCE can not only elevate overall performance but also attain commendable fairness.

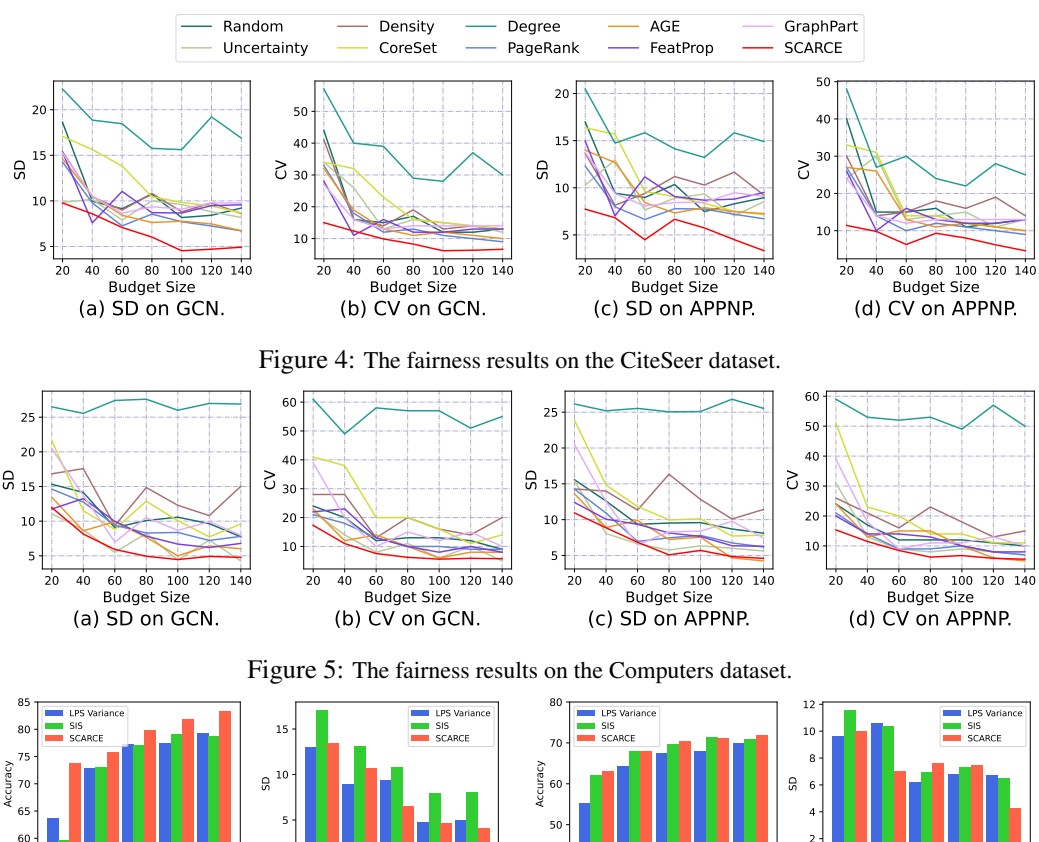

Figure 4: The fairness results on the CiteSeer dataset.

Figure 5: The fairness results on the Computers dataset.

Figure 6: Variants of SCARCE on Cora.

Figure 7: Variants of SCARCE on Citeseer.

# 5 RELATED WORKS

Graph Neural Networks (GNNs) (Kipf & Welling, 2016; Velickovic et al., 2017) have demonstrated significant potential in graph representation learning across a wide range of tasks (Zhou et al., 2020). However, the success of GNNs often depends on the availability of substantial labeled data, which is typically costly and labor-intensive to obtain. Active Learning (AL) methods are designed to address this challenge (Zhan et al., 2022; Ren et al., 2021). AL seeks to strategically select the most informative samples for labeling with the goal of enhancing model accuracy while minimizing the amount of labeled data. Various AL strategies have been proposed, ranging from uncertainty-based methods (Beluch et al., 2018), which pick samples that a model predicts with low confidence, to Representative/Diversity-based approaches (Gal et al., 2017), which emphasize samples that best represent the unlabeled data. While AL has achieved great success with many deep-learning models, its integration with GNNs remains relatively nascent. Works such as Cai et al. (2017); Gao et al. (2018) have attempted to embed node centrality metrics into traditional AL paradigms, but these require iterative sample selection and model retraining, making the process rather inefficient for GNNs. On the other hand, efforts like Wu et al. (2019); Ma et al. (2022) adopt propagated original features for swift node selection, yet their efficacy often hinges on the inherent quality of those features. On the contrary, the proposed SCARCE mainly focuses on the graph structure, which can improve both GNNs performance and fairness.

# 6 CONCLUSION

In this work, we investigate active learning for GNNs, a domain that remains relatively uncharted. Our exploration begins with an evaluation of various criteria, revealing that while original node features may not be informative, criteria based on graph structures are effective indicators of both GNN performance and fairness. We introduce a unified optimization framework for active learning on GNNs. Building on this, we present SCARCE, which is both efficient and can easily integrate with node features. Our comprehensive experiments demonstrate that the proposed SCARCE enhances both the performance and fairness of GNNs.

## 7 ACKNOWLEDGEMENT

This research is supported by the National Science Foundation (NSF) under grant numbers CNS 2246050, IIS1845081, IIS2212032, IIS2212144, IOS2107215, DUE 2234015, DRL 2025244 and IOS2035472, the Army Research Office (ARO) under grant number W911NF-21-1-0198, the Home Depot, Cisco Systems Inc, Amazon Faculty Award, Johnson&Johnson, JP Morgan Faculty Award and SNAP. The work is also supported by MEXT KAKENHI Grant Number 20H04243.

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

# Appendix

## A    PRELIMINARY STUDY

### A.1    EXAMINATION ON ORIGINAL NODE FEATURES

To evaluate the utility of node features in AL, we performed a comparative analysis between the FeatProp method and the Random method using Amazon co-purchase graphs, specifically the Computers and Photo datasets (Shchur et al., 2018). We set our analysis across a budget range of 20 to 200, with accuracy results displayed in Figures 8 and 9. For the Random method, nodes are chosen randomly as labeled nodes, based on the pre-determined budget size. In the case of the FeatProp method, the propagated features are obtained through the operation $\tilde{\mathbf{A}}^2\mathbf{X}$. We then apply the k-means clustering technique, setting the number of clusters equal to the budget size. From each cluster, the node nearest to its center is chosen as the labeled node. Our experimental setup primarily employs two GNN models: GCN (Kipf & Welling, 2016) and APPNP (Klicpera et al., 2018), both used for semi-supervised node classification tasks. The GCN model features two GCN layers with a hidden size of 16, while the APPNP model boasts two linear layers and ten propagation layers, setting its residual ratio at 0.1. Both models use the ReLU activation function. We train all the models 300 epochs and test the model in the final epoch on all nodes in the graph. Our findings suggest that, across most budget sizes, the Random method surpassed FeatProp. This implies that relying solely on original node features may not be beneficial and can, at times, hinder AL performance. As a result, our study's emphasis has been directed towards probing the graph structure.

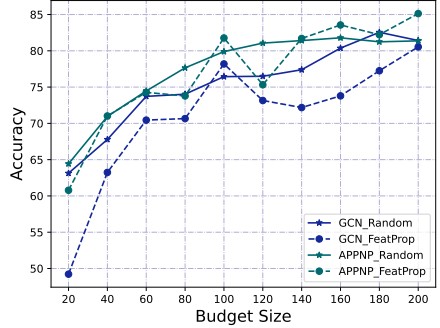

Figure 8: Random vs. FeatProp on Computers.          Figure 9: Random vs. FeatProp on Photo.

### A.2    LABEL POSITION BIAS

In active learning, strategically choosing nodes for labeling can have a significant impact on reducing LPS variance, thereby promoting fairness within GNNs. To empirically investigate the relationship between strategic node selection and GNN performance, we conducted experiments utilizing the Cora and CiteSeer datasets and employed both GCN and APPNP models. In each experiment, we randomly selected 20 nodes for training and performed three runs of GNNs for each selection. Figure 2 and Figure 10 illustrate the correlation between LPS variance and accuracy, based on 200 random training selections for the Cora dataset using GCN and APPNP model, respectively. Similarly, Figure 11 and Figure 12 depict the same relationship observed for the Citeseer dataset. Our findings reveal that: (1) Node selection exerts a profound influence on GNN performance, as exemplified in Figure 2, where we observe an accuracy discrepancy of up to 35%, and in Figure 12, the accuracy discrepancy up to 30%; (2) Selections characterized by lower LPS variance tend to yield superior GNN results. Thus, beyond its role in fostering fairness, minimizing LPS variance emerges as a pivotal criterion for selecting labeled nodes, ultimately enhancing GNN performance.

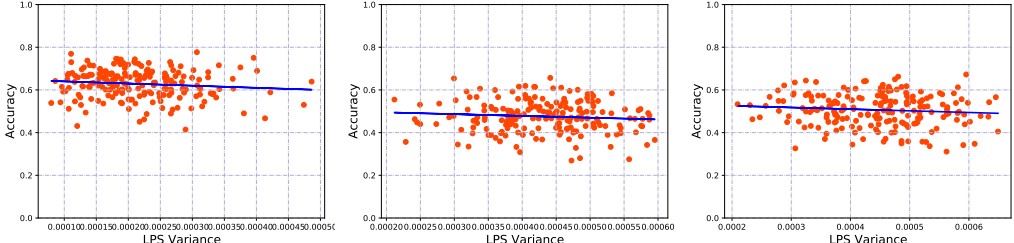

Figure 10: APPNP vs. LPS on Cora.  Figure 11: GCN vs. LPS on CiteSeer.  Figure 12: APPNP vs. LPS on CiteSeer.

### A.3 STRUCTURE INERTIA SCORE

Adhering to the same setting as discussed in A.2, we investigate the interplay between the SIS and the GNNs performance on the Cora and CiteSeer datasets. Figure 3 and Figure 13 shows the the relationship between SIS and accuracy for the Cora dataset, employing GCN and APPNP, respectively, Similarly, Figure 14 and Figure 15 present the corresponding analysis for the Citeseer dataset. Notably, our findings suggest that labeling selections with higher SIS tend to yield superior performance. Therefore, SIS can also function as a structural criterion for selecting labeled nodes.

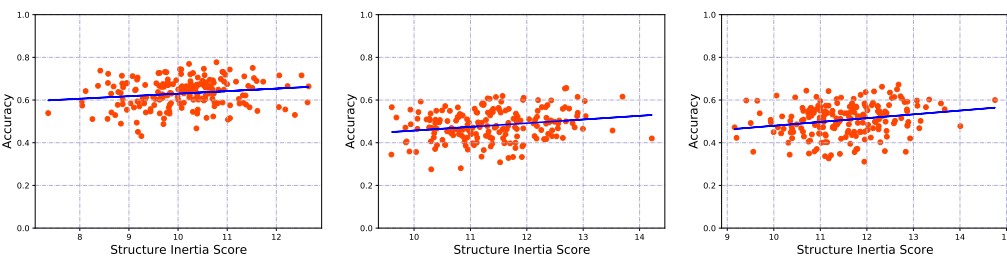

Figure 13: APPNP vs. SIS on Cora.  Figure 14: GCN vs. SIS on CiteSeer.  Figure 15: APPNP vs. SIS on CiteSeer.

## B EXPERIMENT

### B.1 DATASETS STATISTICS

In the experiments, the data statistics used in Section 4 are summarized in Table 3.

Table 3: Dataset Statistics.

| Dataset | Nodes | Edges | Features | Classes | Partition |
|---------|-------|-------|----------|---------|-----------|
| Cora | 2,708 | 5,278 | 1,433 | 7 | 7 |
| CiteSeer | 3,327 | 4,552 | 3,703 | 6 | 14 |
| PubMed | 19,717 | 44,324 | 500 | 3 | 8 |
| Amazon Computer | 13,381 | 245,778 | 767 | 10 | 9 |
| Amazon Photo | 7,487 | 119,043 | 745 | 8 | 5 |
| Ogbn-Arxiv | 169,343 | 1,166,243 | 128 | 40 | 5 |

### B.2 HYPERPARAMETERS SETTING

For the experiments in this paper, we mainly utilize two GNN models: GCN (Kipf & Welling, 2016) and APPNP (Klicpera et al., 2018) for node classification tasks. The GCN model is configured with

two GCN layers, each with a hidden size of 16, while the APPNP model consisted of two linear layers and ten propagation layers, with a residual ratio set to 0.1. ReLU activation functions were employed in both GNN models. For the baseline models, we adhere to the settings in (Ma et al., 2022). We list some fundamental indicators of the baseline models:

| Learning Rate | Dropout Rate | Weight Decay | Hidden Size | Epochs | Activation Function |
|---|---|---|---|---|---|
| 0.01 | 0.5 | 0.0001 | 16 | 300 | ReLU |

Besides, for the GraphPart (Ma et al., 2022) method, a specific partition number is associated with each dataset. In the case of Cora, CiteSeer, PubMed and Ogbn-Arxiv datasets, we adopt the settings outlined in Ma et al. (2022). For Computer and Photo datasets, we also employ the elbow method to determine the appropriate partition number. Specifically, we conduct the k-means clustering on the graph dataset, exploring a range of values for k, typically spanning from 1 to 50. For each value of k, we calculated the sum of squared distances (SSD) between nodes and their respective assigned cluster centers. This metric gauges the proximity of nodes to their cluster centers. Subsequently, we generated a plot that depicts the relationship between the SSD and the number of clusters, k. We identified the elbow point, a juncture where the rate of SSD reduction starts to decelerate. This inflection point signifies that increasing the number of clusters beyond this threshold does not yield a substantial reduction in SSD and might lead to overfitting. The value of k at this elbow point is deemed the optimal number of clusters for the graph dataset, serving as the designated partition number for the dataset.

For the proposed SCARCE, we choose the learning rate $\eta$ from $\{0.001, 0.01, 0.1\}$ based on the gradient scale of t. We set the iteration numbers K = 100, and the number of random trials R = 3000. The code is available at https://anonymous.4open.science/r/SCARCE-D804/.

### B.3 NOISE AND MISSING FEATURE SETTING

In this subsection, we aim to verify the effectiveness of the proposed SCARCE, which can only leverage the graph structure information for active learning, under both noise feature and missing feature settings.

#### B.3.1 NOISE FEATURES

In the noise feature setting, we introduced varying levels of standard Gaussian noise to the original features. Specifically, we modified the features as $X = X + k * \epsilon$, where $\epsilon \sim \mathcal{N}(0, 1)$, and k takes values from the $[0.1, 0.3, 0.5, 0.7, 0.9]$. This experiment was conducted on the Cora and CiteSeer datasets with budget settings of 5C and 10C, where C represents the number of classes in the dataset. We test all the baselines using the GCN and APPNP backbones. The results are shown in Figure 4, Figure 5, Figure 6, and Figure 7, respectively.

Table 4: The performance of GCN model on the Cora dataset with different levels of noise features.

| Budget | 5C | | | | | 10C | | | | |
|---|---|---|---|---|---|---|---|---|---|---|
| Noise | 0.1 | 0.3 | 0.5 | 0.7 | 0.9 | 0.1 | 0.3 | 0.5 | 0.7 | 0.9 |
| Random | 64.52 ± 4.50 | 51.76 ± 4.19 | 44.76 ± 4.12 | 40.54 ± 4.12 | 38.65 ± 3.33 | 74.27 ± 3.32 | 61.78 ± 3.42 | 54.93 ± 2.36 | 51.89 ± 2.18 | 48.41 ± 1.92 |
| Degree | 61.58 ± 0.44 | 54.84 ± 0.66 | 51.25 ± 0.60 | 49.82 ± 0.62 | 47.37 ± 0.79 | 74.67 ± 0.35 | 64.47 ± 0.70 | 59.70 ± 0.54 | 56.69 ± 0.62 | 56.38 ± 0.60 |
| Pagerank | 60.77 ± 1.16 | 43.22 ± 2.59 | 34.96 ± 2.23 | 32.86 ± 0.94 | 32.76 ± 1.27 | 73.22 ± 1.20 | 60.85 ± 1.91 | 50.33 ± 1.65 | 44.68 ± 1.06 | 43.16 ± 1.09 |
| FeatProp | 77.51 ± 0.61 | 45.93 ± 2.18 | 38.50 ± 1.63 | 32.94 ± 0.48 | 26.86 ± 1.45 | 79.48 ± 0.44 | 55.68 ± 1.38 | 52.54 ± 0.89 | 45.03 ± 1.04 | 38.72 ± 0.71 |
| GraphPart | 79.22 ± 0.48 | 62.17 ± 1.24 | 51.80 ± 0.85 | 45.01 ± 1.05 | 40.83 ± 1.56 | 80.24 ± 0.24 | 69.08 ± 1.12 | 58.27 ± 0.89 | 56.24 ± 1.14 | 50.76 ± 1.08 |
| Structure | 75.13 ± 0.81 | 67.23 ± 0.70 | 59.42 ± 0.47 | 55.90 ± 0.63 | 53.45 ± 0.96 | 80.25 ± 0.44 | 71.19 ± 0.61 | 62.82 ± 0.75 | 59.07 ± 1.70 | 55.92 ± 1.52 |

Table 5: The performance of GCN model on the CiteSeer dataset with different levels of noise features.

| Budget | 5C | | | | | 10C | | | | |
|---|---|---|---|---|---|---|---|---|---|---|
| Noise | 0.1 | 0.3 | 0.5 | 0.7 | 0.9 | 0.1 | 0.3 | 0.5 | 0.7 | 0.9 |
| Random | 49.16 ± 8.22 | 36.36 ± 5.71 | 29.53 ± 4.38 | 27.05 ± 2.07 | 26.34 ± 1.85 | 60.43 ± 2.58 | 44.15 ± 2.18 | 37.94 ± 1.87 | 33.94 ± 1.50 | 31.65 ± 1.38 |
| Degree | 44.37 ± 0.88 | 37.27 ± 0.47 | 34.28 ± 0.56 | 34.53 ± 0.46 | 34.12 ± 0.56 | 46.20 ± 0.80 | 39.50 ± 0.91 | 36.71 ± 0.55 | 35.54 ± 0.54 | 35.55 ± 0.53 |
| Pagerank | 49.34 ± 2.77 | 33.40 ± 2.49 | 27.99 ± 1.22 | 25.41 ± 1.95 | 25.08 ± 1.74 | 61.03 ± 0.93 | 41.22 ± 1.27 | 34.20 ± 0.96 | 32.94 ± 0.66 | 30.82 ± 1.16 |
| FeatProp | 52.30 ± 2.47 | 33.16 ± 1.62 | 27.59 ± 1.84 | 26.10 ± 1.25 | 24.88 ± 0.98 | 64.48 ± 1.17 | 36.84 ± 2.16 | 28.13 ± 1.12 | 25.03 ± 1.11 | 25.59 ± 1.08 |
| GraphPart | 52.37 ± 1.13 | 44.63 ± 1.05 | 38.91 ± 0.66 | 35.59 ± 0.92 | 34.42 ± 1.51 | 59.19 ± 1.78 | 49.32 ± 1.18 | 40.48 ± 1.13 | 37.61 ± 0.84 | 35.45 ± 1.17 |
| Structure | 59.50 ± 2.53 | 50.35 ± 0.56 | 41.61 ± 0.83 | 37.23 ± 0.63 | 35.23 ± 1.11 | 68.23 ± 0.14 | 53.54 ± 0.53 | 45.34 ± 0.98 | 41.88 ± 0.99 | 40.09 ± 0.67 |

From these results, we can have the following observations:

Table 6: The performance of APPNP model on the Cora dataset with different levels of noise features.

| Budget | 5C | | | | | 10C | | | | |
|---|---|---|---|---|---|---|---|---|---|---|
| Noise | 0.1 | 0.3 | 0.5 | 0.7 | 0.9 | 0.1 | 0.3 | 0.5 | 0.7 | 0.9 |
| Random | 69.52 ± 5.14 | 60.06 ± 4.43 | 55.21 ± 4.19 | 53.16 ± 3.60 | 51.51 ± 4.42 | 77.09 ± 3.86 | 69.81 ± 3.45 | 64.39 ± 2.65 | 62.38 ± 2.99 | 60.89 ± 2.35 |
| Degree | 67.59 ± 0.92 | 61.86 ± 0.55 | 58.98 ± 0.67 | 56.29 ± 0.82 | 55.81 ± 0.88 | 75.67 ± 0.85 | 70.38 ± 0.93 | 66.39 ± 0.41 | 63.33 ± 0.94 | 63.49 ± 0.66 |
| Pagerank | 70.87 ± 1.70 | 54.90 ± 2.85 | 50.41 ± 1.87 | 44.75 ± 1.93 | 43.62 ± 2.49 | 76.23 ± 0.90 | 67.79 ± 1.72 | 64.51 ± 1.81 | 60.13 ± 1.28 | 58.05 ± 1.62 |
| FeatProp | 80.17 ± 0.77 | 48.23 ± 1.72 | 44.64 ± 1.36 | 35.49 ± 0.77 | 35.76 ± 0.86 | 81.05 ± 0.55 | 61.54 ± 1.19 | 53.96 ± 0.73 | 45.31 ± 1.91 | 39.97 ± 1.04 |
| GraphPart | 80.68 ± 0.41 | 68.76 ± 1.80 | 66.11 ± 1.05 | 59.14 ± 2.04 | 54.07 ± 2.22 | 84.05 ± 0.28 | 73.79 ± 1.02 | 66.61 ± 0.59 | 63.17 ± 1.10 | 61.91 ± 1.19 |
| Structure | 79.39 ± 0.67 | 75.89 ± 0.66 | 72.46 ± 0.20 | 70.10 ± 1.21 | 69.19 ± 0.78 | 82.63 ± 0.59 | 75.73 ± 1.57 | 68.73 ± 1.87 | 66.90 ± 1.73 | 66.08 ± 2.38 |

Table 7: The performance of APPNP model on the CiteSeer dataset with different levels of noise features.

| Budget | 5C | | | | | 10C | | | | |
|---|---|---|---|---|---|---|---|---|---|---|
| Noise | 0.1 | 0.3 | 0.5 | 0.7 | 0.9 | 0.1 | 0.3 | 0.5 | 0.7 | 0.9 |
| Random | 50.91 ± 7.05 | 39.35 ± 4.37 | 35.08 ± 5.96 | 32.97 ± 3.49 | 31.77 ± 4.15 | 62.32 ± 1.88 | 47.88 ± 2.36 | 43.08 ± 2.42 | 39.47 ± 1.83 | 38.86 ± 2.07 |
| Degree | 50.83 ± 1.58 | 41.98 ± 1.10 | 39.77 ± 0.62 | 36.92 ± 0.38 | 36.84 ± 0.51 | 50.81 ± 2.35 | 43.26 ± 1.00 | 40.88 ± 0.99 | 38.80 ± 0.69 | 37.82 ± 0.81 |
| Pagerank | 52.08 ± 2.55 | 41.20 ± 2.01 | 34.63 ± 2.43 | 30.68 ± 2.27 | 31.78 ± 1.99 | 59.43 ± 1.23 | 46.77 ± 1.40 | 41.61 ± 1.01 | 37.87 ± 0.79 | 35.63 ± 0.85 |
| FeatProp | 54.49 ± 1.66 | 36.12 ± 1.34 | 27.69 ± 1.57 | 25.43 ± 0.81 | 24.31 ± 1.53 | 65.09 ± 1.02 | 37.86 ± 0.87 | 32.11 ± 1.05 | 34.98 ± 0.98 | 27.70 ± 1.05 |
| GraphPart | 60.38 ± 1.40 | 45.39 ± 1.36 | 40.97 ± 1.14 | 39.19 ± 1.78 | 38.79 ± 1.11 | 61.59 ± 0.98 | 49.92 ± 1.00 | 42.31 ± 1.26 | 41.18 ± 1.12 | 39.40 ± 1.12 |
| Structure | 62.58 ± 1.24 | 54.76 ± 0.82 | 48.60 ± 0.63 | 45.67 ± 0.55 | 44.28 ± 0.60 | 69.28 ± 0.22 | 59.10 ± 2.33 | 47.34 ± 1.73 | 44.65 ± 2.12 | 43.55 ± 1.59 |

- With the level of noise increase in the features, the performance of the feature-based methods drops a lot. For example, the FeatProp performs even worse than random selection.

- The proposed SCARCE, which only leverages the graph structure in active learning, can outperform baselines by a large margin.

### B.3.2 MISSING FEATURES

To further test the resilience of SCARCE under challenging feature conditions, we conducted experiments in a setting where all features are missing. In this scenario, we replaced the original node features with one-hot ID features. We conduct experiments on both Cora and CiteSeer datasets with labeling budgets of 5C, 10C, and 20C using both GCN and APPNP models. The results are shown in Figure 8 and Figure 9, respectively.

Table 8: The performance of GCN model on Cora and CiteSeer dataset without node features.

| Dataset | Cora | | | CiteSeer | | |
|---|---|---|---|---|---|---|
| Budget | 5C | 10C | 20C | 5C | 10C | 20C |
| Random | 44.76 ± 5.83 | 56.47 ± 2.30 | 64.03 ± 1.74 | 28.41 ± 2.69 | 34.78 ± 3.06 | 40.58 ± 2.61 |
| Degree | 44.19 ± 7.98 | 51.69 ± 2.82 | 63.26 ± 1.95 | 27.33 ± 1.97 | 31.74 ± 2.79 | 40.99 ± 2.02 |
| Pagerank | 39.19 ± 3.55 | 47.95 ± 3.94 | 63.93 ± 3.85 | 24.03 ± 1.96 | 32.83 ± 2.21 | 40.35 ± 2.02 |
| FeatProp | 47.75 ± 3.16 | 58.84 ± 3.22 | 65.66 ± 1.38 | 29.79 ± 2.32 | 34.79 ± 2.00 | 41.89 ± 2.71 |
| GraphPart | 55.37 ± 4.06 | 57.40 ± 4.24 | 67.60 ± 2.61 | 38.27 ± 2.08 | 37.40 ± 2.56 | 43.61 ± 3.15 |
| SCARCE | 61.43 ± 2.31 | 68.64 ± 1.94 | 69.59 ± 1.22 | 37.89 ± 1.94 | 43.51 ± 1.44 | 48.20 ± 1.15 |

Table 9: The performance of APPNP model on Cora and CiteSeer dataset without node features.

| Dataset | Cora | | | CiteSeer | | |
|---|---|---|---|---|---|---|
| Budget | 5C | 10C | 20C | 5C | 10C | 20C |
| Random | 62.27 ± 3.86 | 68.71 ± 3.22 | 75.52 ± 1.65 | 38.24 ± 3.21 | 46.10 ± 2.85 | 49.98 ± 1.52 |
| Degree | 64.48 ± 2.97 | 65.29 ± 1.63 | 73.21 ± 1.17 | 38.92 ± 4.79 | 46.93 ± 1.60 | 49.10 ± 2.16 |
| Pagerank | 61.79 ± 1.61 | 68.89 ± 1.11 | 74.15 ± 0.58 | 31.94 ± 1.20 | 42.58 ± 1.39 | 50.70 ± 1.05 |
| FeatProp | 35.22 ± 0.99 | 47.42 ± 3.34 | 72.47 ± 0.42 | 31.89 ± 0.46 | 31.98 ± 0.57 | 29.01 ± 0.56 |
| GraphPart | 66.40 ± 0.60 | 71.98 ± 0.59 | 74.15 ± 1.38 | 46.10 ± 0.62 | 49.93 ± 0.40 | 51.34 ± 0.78 |
| SCARCE | 72.39 ± 1.54 | 76.33 ± 0.99 | 76.77 ± 0.71 | 48.06 ± 3.66 | 52.20 ± 2.02 | 56.21 ± 1.92 |

From the results, we can have similar findings: While the feature-based methods, such as FeatProp, perform poorly under the missing feature setting, our proposed SCARCE can have good performance.

In summary, the proposed method SCARCE can perform as expected when the feature quality is low or missing, but the feature-based active learning methods may perform poorly.

## B.4 The Performance with More Backbones

To verify the effectiveness of the proposed SCARCE with different backbones. We further add two representative GNNs: one is GAT Velickovic et al. (2017), which leverages the attention mechanism, and the other one is the GCNII Chen et al. (2020), which is a deep GNN. For these two methods, we follow the hyperparameter settings in their original paper. Specifically, for GCNII, we adopt 64 layers. We test these two backbones using the Cora and CisteSeer datasets with budgets of 5C, 10C, and 20C. The results with GCNII and GAT backbones are shown in Figure 10 and Figure 11, respectively. From these results, we can observe that the proposed SCARCE can work well with both GAT and GCNII backbones.

Table 10: The performance on Cora and CiteSeer dataset with GCNII backbone.

| Dataset | Cora | | | CiteSeer | | |
|---|---|---|---|---|---|---|
| Budget | 5C | 10C | 20C | 5C | 10C | 20C |
| Random | 69.76 ± 6.47 | 78.29 ± 3.99 | 83.24 ± 1.08 | 55.25 ± 7.68 | 65.59 ± 1.60 | 68.25 ± 1.28 |
| Uncertainty | 68.78 ± 5.64 | 77.15 ± 3.36 | 82.33 ± 1.30 | 53.59 ± 8.70 | 62.91 ± 5.47 | 67.97 ± 1.69 |
| Density | 72.31 ± 3.10 | 78.85 ± 1.61 | 83.22 ± 1.08 | 55.49 ± 7.04 | 65.41 ± 2.80 | 68.07 ± 1.51 |
| CoreSet | 66.07 ± 3.34 | 76.11 ± 3.37 | 80.43 ± 1.97 | 54.22 ± 6.17 | 61.45 ± 3.05 | 67.41 ± 1.02 |
| Degree | 68.02 ± 1.92 | 77.92 ± 0.45 | 82.85 ± 0.41 | 58.99 ± 1.61 | 60.62 ± 1.95 | 67.41 ± 0.50 |
| Pagerank | 69.82 ± 2.74 | 76.64 ± 1.27 | 83.52 ± 0.51 | 53.41 ± 2.70 | 62.61 ± 1.00 | 69.53 ± 0.66 |
| AGE | 71.31 ± 3.89 | 78.81 ± 2.14 | 83.29 ± 1.23 | 56.65 ± 4.73 | 64.46 ± 2.58 | 67.59 ± 2.60 |
| FeatProp | 74.63 ± 2.59 | 83.85 ± 0.58 | 84.49 ± 0.60 | 44.99 ± 2.25 | 52.97 ± 2.08 | 63.01 ± 0.93 |
| GraphPart | 80.45 ± 1.07 | 82.47 ± 0.46 | 83.4 ± 0.42 | 47.89 ± 2.25 | 58.99 ± 1.34 | 65.99 ± 1.08 |
| Structure | 79.96 ± 1.10 | 83.43 ± 0.54 | 83.98 ± 0.48 | 64.55 ± 0.72 | 69.99 ± 0.74 | 72.05 ± 0.35 |
| Feature | 82.05 ± 1.32 | 83.25 ± 0.65 | 85.96 ± 0.53 | 67.63 ± 1.98 | 70.98 ± 0.37 | 72.25 ± 0.74 |
| ALL | 82.61 ± 1.01 | 83.15 ± 0.47 | 85.94 ± 0.51 | 66.37 ± 4.07 | 70.91 ± 0.42 | 72.89 ± 0.44 |

Table 11: The performance on Cora and CiteSeer dataset with GAT backbone.

| Dataset | Cora | | | CiteSeer | | |
|---|---|---|---|---|---|---|
| Budget | 5C | 10C | 20C | 5C | 10C | 20C |
| Random | 65.66 ± 3.70 | 74.06 ± 4.19 | 80.00 ± 2.24 | 51.36 ± 7.53 | 61.85 ± 2.34 | 64.82 ± 2.15 |
| Uncertainty | 55.31 ± 5.78 | 62.02 ± 6.56 | 74.72 ± 3.69 | 44.41 ± 6.35 | 49.95 ± 5.64 | 60.49 ± 4.00 |
| Density | 64.10 ± 4.28 | 70.72 ± 3.73 | 78.12 ± 2.97 | 49.44 ± 7.58 | 60.10 ± 3.73 | 64.17 ± 2.02 |
| CoreSet | 60.72 ± 7.03 | 72.54 ± 3.40 | 79.48 ± 1.52 | 47.29 ± 6.76 | 58.00 ± 4.21 | 64.54 ± 2.11 |
| Degree | 62.34 ± 2.20 | 73.44 ± 1.39 | 78.31 ± 0.98 | 42.18 ± 1.89 | 44.41 ± 1.29 | 49.38 ± 2.20 |
| Pagerank | 63.78 ± 2.05 | 71.84 ± 1.77 | 80.00 ± 1.03 | 42.98 ± 2.40 | 58.62 ± 1.87 | 66.51 ± 0.77 |
| AGE | 66.45 ± 4.71 | 73.93 ± 3.89 | 80.14 ± 1.55 | 49.53 ± 5.40 | 58.97 ± 5.94 | 65.18 ± 2.26 |
| FeatProp | 73.40 ± 1.39 | 77.62 ± 1.36 | 82.36 ± 0.53 | 58.03 ± 2.99 | 62.14 ± 2.29 | 68.08 ± 0.54 |
| GraphPart | 76.91 ± 1.41 | 78.29 ± 0.88 | 80.94 ± 1.52 | 58.24 ± 2.41 | 65.10 ± 1.32 | 66.41 ± 1.33 |
| Structure | 76.36 ± 2.00 | 81.78 ± 0.46 | 82.58 ± 0.41 | 64.23 ± 1.92 | 69.32 ± 0.60 | 72.36 ± 0.49 |
| Feature | 80.56 ± 0.77 | 83.01 ± 0.63 | 85.04 ± 0.26 | 60.91 ± 3.10 | 70.47 ± 0.61 | 72.60 ± 0.38 |
| ALL | 80.62 ± 0.50 | 83.27 ± 0.39 | 84.96 ± 0.24 | 61.20 ± 2.77 | 70.29 ± 0.61 | 72.34 ± 0.40 |

## B.5 The performance on OGB Products dataset

To verify the scalability of the proposed SCARCE, we further conducted extensive experiments on the OGB Products dataset, which contains over 2.4 million nodes. We also use the budgets of 5C, 10C, and 20C. The results with both GCN and APPNP backbones are shown in Figure 12

Table 12: Products

| Dataset | GCN | | | APPNP | | |
|---|---|---|---|---|---|---|
| Budget | 5C | 10C | 20C | 5C | 10C | 20C |
| Random | 60.78 ± 0.60 | 67.38 ± 0.74 | 70.45 ± 0.89 | 66.15 ± 1.43 | 70.23 ± 0.42 | 74.82 ± 0.10 |
| Degree | 37.13 ± 0.46 | 45.82 ± 0.38 | 50.19 ± 0.99 | 49.98 ± 0.85 | 55.43 ± 0.74 | 60.81 ± 0.97 |
| Pagerank | 56.33 ± 0.60 | 63.20 ± 0.20 | 69.00 ± 0.45 | 61.08 ± 0.32 | 67.61 ± 0.70 | 72.69 ± 0.18 |
| FeatProp | 63.34 ± 2.58 | 67.48 ± 0.97 | 71.65 ± 0.57 | 68.64 ± 0.65 | 69.80 ± 1.20 | 71.8 ± 0.61 |
| GraphPart | OOT | OOT | OOT | OOT | OOT | OOT |
| SCARCE | 62.25 ± 0.37 | 70.27 ± 0.22 | 74.43 ± 0.22 | 66.66 ± 0.09 | 72.63 ± 0.20 | 76.57 ± 0.14 |

Please note the GraphPart method takes more than 24 hours, and we treat it as out of time (OOT).

The results from our experiments on the large-scale OGB Products dataset clearly demonstrate the efficacy of our proposed SCARCE method. Notably, SCARCE achieves high accuracy with significantly fewer labeled nodes compared to standard training splits. For instance, while the standard split typically uses around 200,000 nodes for training, SCARCE requires only 940 labeled nodes to attain accuracies of 74.43/(75.64) and 76.57/(76.62) for the GCN and APPNP models, respectively, where the value in the $(\cdot)$ represent the accuracy of the standard split. This underscores SCARCE's effectiveness in active learning, particularly in large-scale graph neural network applications.

