# OpenReview forum: "Structural Fairness-aware Active Learning for Graph Neural Networks"
_ICLR.cc/2024/Conference — ICLR 2024 poster_

### Official Review · Reviewer_RKZS · 2023-10-23

**Soundness:** 2 fair
**Presentation:** 2 fair
**Contribution:** 2 fair
**Rating:** 5
**Confidence:** 3

**Summary:**

The study focuses on enhancing the performance of GNNs for semi-supervised node classification, even when high-quality labeled samples are scarce. Traditional active learning methods may not work optimally in graph data, given their unique structures and the bias introduced by the positioning of labeled nodes. To address this, the researchers introduce a unified optimization framework called SCARCE, which can be combined with node features. Their experiments confirm that this method not only enhances GNN performance but also helps mitigate structural bias and improve fairness in the results.

**Strengths:**

1. Comparison with many baselines in this paper is very good.

**Weaknesses:**

1. The paper is not easy to follow. I suggest the existing work on fairness in GNN should be discussed. Otherwise, it is very hard to estimate the significance of this work.

2. My major concern is that the fairness definition in this paper is not very clear. We usually use demographic party (DP) or equal odds (EO) to measure fairness. However, this paper only uses the Standard Deviation (SD) and the Coefficient of Variation. Why do the authors consider them instead of DP and EO?

3. Figure 6 and Figure 7 are also not very clear to me. The authors discussed that 'SCARCE, which combines both SIS and LPS variance, SCARCE can not only elevate overall performance but also attain commendable fairness'. However, it is very hard to get this result from these two figures. I suggest the authors provide more details for examination.

4. This paper should focus on fairness instead of classification accuracy. However, Tables 1 and 2 provide more details about the classification accuracy. There should be a trade-off between accuracy and fairness. Only showing accuracy does not make any sense. In addition, how to balance the trade-off between accuracy and fairness in this paper. I do not find any implementation details related to this.

**Questions:**

See Weakness.

---

> ### Author Response · Authors · 2023-11-18
> **Response to Reviewer RKZS - Part 1**
>
> Dear Reviewer RKZS,
>
> We appreciate your constructive feedback. We are pleased to provide detailed responses to address your concerns.
>
> > **Q1**: I suggest the existing work on fairness in GNN should be discussed. Otherwise, it is very hard to estimate the significance of this work.
>
> **R1**: Thank you for your suggestions. It is important to clarify that the type of fairness our paper addresses differs significantly from the commonly studied attribute bias, which pertains to model performance disparity across different sensitive attribute groups, such as gender and race.
>
> Our work concentrates on structural fairness, specifically tackling the issue of "Label Position Bias" in GNNs.
> This bias emerges due to the varying performance of a GNN model for nodes based on their "structural distance" to labeled nodes in the graph. Research [1] uses the Label Proximal Score (LPS) to quantify this proximity, with a higher LPS indicating a closer "distance" to labeled nodes and typically correlating with better model performance. They found the low variance of the LPS score, which means all the nodes have a similar "distance" to the labeled node, can mitigate the label position bias and improve the performance fairness of GNNs. Thus, we use the LPS variance to measure structural fairness. One natural way of reducing the LPS variance is to strategically select the labeled nodes based on their position in the graph.
>
> The goal of active learning is to strategically select some nodes to label that can maximize the model's performance.
> Therefore, both active learning and mitigating the label position bias are fundamentally linked to the labeling of nodes in a graph. Our preliminary studies illustrate that structure fairness (Low LPS variance) is also related to the overall performance of GNNs, which is the central focus of active learning. Therefore, in this work, we use the LPS variance as one metric for active learning to select nodes to label. The proposed method not only improves performance but also addresses the fairness issue.
>
> As for the existing works on fairness in GNNs, most of them focus on attribute fairness, which is not related to the selection of labeled nodes. Therefore, they are not related to labeling, which is the target of active learning. The pioneering work [1] identifies label position bias and proposes solving it by learning an unbiased graph structure. Our contribution differs as we approach the issue from a labeling perspective, offering an alternative solution to the problem.
>
> Additionally, our method has the advantage of performing well even when node features are noisy or missing, as shown in the experiments in Appendix Section B.3 of our revised paper. For your convenience, we also present partial results here:
>
> 1. Noise feature setting: we add different levels of gaussion noise to the node features.
>
> |GCN|||Cora|||||CiteSeer|||
> |:---:|:---:|:---:|:---:|:---:|:---:|:---:|:---:|:---:|:---:|:---:|
> |Noise|0.1|0.3|0.5|0.7|0.9|0.1|0.3|0.5|0.7|0.9|
> |Random|64.52±4.50|51.76±4.19|44.76±4.12|40.54±4.12|38.65±3.33|49.16±8.22|36.36±5.71|29.53±4.38|27.05±2.07|26.34±1.85|
> |Degree|61.58±0.44|54.84±0.66|51.25±0.60|49.82±0.62|47.37±0.79|44.37±0.88|37.27±0.47|34.28±0.56|34.53±0.46|34.12±0.56|
> |Pagerank|60.77±1.16|43.22±2.59|34.96±2.23|32.86±0.94|32.76±1.27|49.34±2.77|33.40±2.49|27.99±1.22|25.41±1.95|25.08±1.74|
> |FeatProp|77.51±0.61|45.93±2.18|38.50±1.63|32.94±0.48|26.86±1.45|52.30±2.47|33.16±1.62|27.59±1.84|26.10±1.25|24.88±0.98|
> |GraphPart|79.22±0.48|62.17±1.24|51.80±0.85|45.01±1.05|40.83±1.56|52.37±1.13|44.63±1.05|38.91±0.66|35.59±0.92|34.42±1.51|
> |SCARCE|75.13±0.81|67.23±0.70|59.42±0.47|55.90±0.63|53.45±0.96|59.50±2.53|50.35±0.56|41.61±0.83|37.23±0.63|35.23±1.11|
>
> 2. Missing feature setting: we use the one-hot ID as node features.
> |GCN||Cora|||CiteSeer||
> |:---:|:---:|:---:|:---:|:---:|:---:|:---:|
> |Budget|5C|10C|20C|5C|10C|20C|
> |Random|44.76±5.83|56.47±2.30|64.03±1.74|28.41±2.69|34.78±3.06|40.58±2.61|
> |Degree|44.19±7.98|51.69±2.82|63.26±1.95|27.33±1.97|31.74±2.79|40.99±2.02|
> |Pagerank|39.19±3.55|47.95±3.94|63.93±3.85|24.03±1.96|32.83±2.21|40.35±2.02|
> |FeatProp|47.75±3.16|58.84±3.22|65.66±1.38|29.79±2.32|34.79±2.00|41.89±2.71|
> |GraphPart|55.37±4.06|57.40±4.24|67.60±2.61|38.27±2.08|37.40±2.56|43.61±3.15|
> |SCARCE|61.43±2.31|68.64±1.94|69.59±1.22|37.89±1.94|43.51±1.44|48.20±1.15|
>
> From the above results, we can find that the feature-based active learning method, such as FeatProp, can't perform well when the noise level is high or all the features are missed. However, our proposed SCARCE can only leverage the graph structure and perform well on both scenarios.
>
> [1] Towards Label Position Bias in Graph Neural Networks, NeurIPS'23

---

> ### Author Response · Authors · 2023-11-18
> **Response to Reviewer RKZS - Part 2**
>
> > **Q2**: My major concern is that the fairness definition in this paper is not very clear. We usually use demographic party (DP) or equal odds (EO) to measure fairness. However, this paper only uses the Standard Deviation (SD) and the Coefficient of Variation. Why do the authors consider them instead of DP and EO?
>
> **R2**: The performance fairness issue that we focused on in this paper is caused by the label position bias. We would like to first provide the definitions:
>
> **Label Position Bias & Structural Fairness**: In this paper, we use the notion of  "Label Position Bias", a critical aspect of performance bias in GNNs. This bias arises when the performance of a GNN model varies for nodes based on their proximity to labeled nodes in the graph.[1] Formally, in a graph G, with the set of labeled nodes $V_L$, and unlabeled nodes $V_U$, a GNN model trained on $V_L$ may exhibit varying performance for two nodes $i, j \in V_U $ based on their "structural distance" to $V_L$. A key metric we use to measure this distance is the Label Proximal Score $LPS_{i} = \sum_{j \in V_L} P_{ij}$, where P is the personalized PageRank matrix. The LPS can well quantify the proximity of unlabeled nodes to labeled nodes and a higher LPS score typically correlates with better model performance for that node [1]. [1] also found the low variance of the LPS score, which indicates all the nodes have a similar "distance" to the labeled nodes, can mitigate the label position bias and improve the performance fairness of GNNs. As a result, we use the variance of the LPS score of each node as the measure of structural fairness.
>
> **Measuring Performance Fairness**: To quantify performance fairness, we categorize nodes into different sensitive groups based on their LPS scores. For each group, we calculate the average accuracy of the GNNs and use metrics such as Standard Deviation to measure the performance disparity across these groups.
>
> In fact, when considering multiple sensitive groups, the concepts of DP and SD are the same.  If there are only two sensitive groups $i$ and $j$ with accuracy $a_i$ and $a_j$, the $DP = |a_i - a_j|$. However, if there are multiple sensitive groups, $DP = \sum_i |a_i - \bar{a}|$, where $\bar{a}$ is the average accuracy. Therefore, we use the Standard Deviation $SD = \sqrt{\sum_i (a_i - \bar{a})^2}$ to measure the fairness.
> In our paper, we use the Label Proximity Score to split the nodes into multiple sensitive groups. This realization aligns with your suggestion, and we will update our terminology about SD to reflect this more accurately in our revised paper. Additionally, we use the Coefficient of Variation (CV) because it normalizes the standard deviation by the mean, $CV = SD / \bar{a}$. This normalization is particularly useful for comparing the relative disparity in performance across models with different overall accuracy levels, allowing us to assess fairness in a way that accounts for differences in model performance.
>
> > **Q3**: Figure 6 and Figure 7 are also not very clear to me. The authors discussed that `SCARCE, which combines both SIS and LPS variance, SCARCE can not only elevate overall performance but also attain commendable fairness'. However, it is very hard to get this result from these two figures. I suggest the authors provide more details for examination.
>
> **R3**:  Thank you for your feedback on Figures 6 and 7, which might be uneasy to read. The proposed SCARCE combines two metrics, i.e., the LPS variance and SIS, to strategically select nodes for annotation. Figures 6 and 7 showcase the impact of these metrics on the Cora and CiteSeer datasets, respectively. For each figure, the left subfigure represents the accuracy, and the right subfigure represents the fairness (Standard Deviation) of using different metrics. From Figure 6 and 7, we can find:
> * For accuracy, the proposed SCARCE, which combines both LPS variance and SIS, consistently achieves better accuracy compared to using either metric alone.
> * For fairness, using the LPS variance alone tends to cause a low SD (better fairness). This finding is consistent with our motivation for leveraging the LPS variance to address the fairness issue.  The SIS alone may lead to a higher SD. However, when combining these two metrics in SCARCE, we observe that the model maintains commendable fairness (lower SD).

---

> ### Author Response · Authors · 2023-11-18
> **Response to Reviewer RKZS - Part 3**
>
> > **Q4**: This paper should focus on fairness instead of classification accuracy. However, Tables 1 and 2 provide more details about the classification accuracy. There should be a trade-off between accuracy and fairness. Only showing accuracy does not make any sense. In addition, how to balance the trade-off between accuracy and fairness in this paper. I do not find any implementation details related to this.
>
> **R4**: We would like to clarify that the main focus of this paper is the active learning for the graph neural networks, which aims to maximize the model's performance within a labeling budget. We propose a unified framework that effectively combines different metrics in active learning. As discussed in **R1**, both active learning and mitigating label position bias are fundamentally related to the labeling of nodes in a graph. Besides, structure fairness (low LPS variance) is also related to the overall performance of GNNs, as demonstrated in the preliminary study. Therefore, we leverage structural fairness as one metric in active learning to maximize GNNs' performance.
> Meanwhile, we found this metric can also improve performance fairness, as shown in Figures 4 & 5 in our paper. Therefore, it is not a trade-off between accuracy and fairness. As demonstrated in Figures 6 & 7, incorporating the LPS variance can improve the model's overall performance and performance fairness. As a result, the proposed method bridges the gap between active learning and mitigating the label position bias in GNNs, proposing that addressing fairness can be synergistic with achieving high model performance.
>
>
> We hope that we have addressed the concerns in your comments, and please kindly let us know if there is any further concern, and we are happy to clarify.

---

> ### Author Response · Authors · 2023-11-20
> **A Gentle Remind to Reviewer RKZS**
>
> Dear Reviewer RKZS,
>
> We would like to express our sincere gratitude to you for reviewing our paper and providing valuable feedback. Could we kindly know if the responses have addressed your concerns? If there are any further questions, we are happy to clarify. Thank you.
>
> Best,
>
> All authors

---

> > ### Comment · Reviewer_RKZS · 2023-11-20
> > **Response**
> >
> > Dear Authors,
> >
> > Thank you for your response. While I appreciate your elaboration on the contributions and objectives of the paper, I believe the emphasis on fairness may not be entirely appropriate in this paper. It appears that active learning is the primary focus, yet the definitions and applications of fairness within the paper lack clarity.
> >
> > Most importantly,  it's worth noting that this concern is not unique to me. I checked other reviewer's comments. Reviewer UJDy also has the same concern.
> >
> > Given this, my recommendation would be to revise the paper with a reduced emphasis on fairness, possibly even removing the term from the title. This is merely a suggestion, and I understand it's ultimately at your discretion.
> >
> > After reviewing your response, which addressed some of my questions, I am inclined to adjust my evaluation. I will raise my score from 3 to 5.
> >
> > Thanks.
> >
> > Reviewer  RKZS.

---

> > > ### Author Response · Authors · 2023-11-20
> > > **Further Response to Reviewer RKZS**
> > >
> > > Dear Reviewer RKZS,
> > >
> > > Thank you for your prompt reply and great suggestion. We are glad to know that our previous responses have addressed most of your concerns, and we are happy to address your remaining question: the term of fairness.
> > >
> > > We agree that the current wording on 'fairness' may lead to ambiguity, and we appreciate your guidance in enhancing the clarity and impact of our work. The fairness discussed in this paper is about a very new concept recently proposed, namely "label position bias'' (or "bias'' for short once we make the context clear), which is unique for graphs and quite different from the commonly discussed fairness, which confuses you and another reviewer. Therefore, to fully address this ambiguity, we would like to change this wording to "label positional bias'' for better clarity.
> > >
> > > Furthermore, we propose the following revision plan to remove the ambiguity you are concerned about:
> > >
> > > 1. Terminology: Throughout this paper, we will change the "fairness" to "bias", specifically for the label position bias, which is caused by the labeling of nodes in graphs. Mitigating the label position bias is well aligned with the goal of active learning - labeling.
> > >
> > > 2. Title: We will change the title to "Structural Bias-Aware Active Learning for Graph Neural Networks".
> > >
> > > 3. Introduction: In paragraph 3, we will introduce more about label position bias, such as the problems caused by it. For example, in real-world applications like fraud detection, where labeled data might be scarce, users far away from these labeled nodes are at a higher risk of being misclassified due to the label position bias. Then, we will point out that unbiased labeling can mitigate the label position bias and improve overall performance as demonstrated in the preliminary study. This builds the connection between active learning and mitigating the label position bias.
> > >
> > > 4. Preliminary: In section 2.2, we will define unbiased labeling as node labeling with small LPS variance. Through the preliminary studies, we demonstrate that unbiased labeling can improve the overall performance while mitigating the label position bias. As a result, we can use it as one of the structural criteria in active learning.
> > >
> > > 5. Experiments: Section 4.3 will be renamed from "Fairness Comparison" to "Evaluating Bias Mitigation Performance" to reduce ambiguity;
> > > In Section 4.4, we will introduce experiments under noise and missing feature scenarios to demonstrate the effectiveness of our proposed structural criteria.
> > >
> > > These revisions will significantly diminish any ambiguity associated with the term 'fairness' and establish a clearer relationship between active learning and the mitigation of label position bias in graph neural networks. Moreover, the proposed revisions are minor and should be easily accomplishable while mitigating confusion.
> > >
> > > We hope these changes meet your expectations and contribute to a more precise and impactful paper. If you are satisfied with the revision plan, we will upload the revised version. If you have any further questions or suggestions, we are happy to discuss them with you.
> > >
> > > Best Regards,
> > >
> > > All Authors

---

### Official Review · Reviewer_tqC2 · 2023-10-31

**Soundness:** 1 poor
**Presentation:** 2 fair
**Contribution:** 1 poor
**Rating:** 6
**Confidence:** 3

**Summary:**

To leverage graph structure and mitigate structural bias in active learning, the authors present a unified optimization framework.

**Strengths:**

Originality: The investigation of structure fairness using active learning is something new.

Quality: The technical quality is below average. Many details are missing. For example, label position bias is a new term that was recently proposed, and the authors should elaborate more on it with a more intuitive explanation instead of just some formula. Also, at the beginning of Sec. 3.1, $t$ is a binary vector, and thus it should be $t \in \\{0,1\\}^n$. The relaxation in the paper does not make sense to the reviewer.

Clarity: In general, it is ok. The reviewer understands how the proposed method works but sometimes fails to see why.

Significance: The fairness and active learning problems for graphs are important.

**Weaknesses:**

1. The paper tries to solve the structure fairness problem using active learning. However, the connection between these two is weak, and the reviewer does not find any strong motivations to do so. The authors claim that "in active learning, strategically choosing labeling nodes, represented by t, can potentially reduce the LPS variance, promoting fairness in GNNs" in the second para. of Sec. 2.2, but the reviewer does not find any theoretical guarantees to motivate this finding.

2. The goal of active learning is different from mitigating the bias in graphs and the ultimate goal of AL is to use as few as labeled nodes to achieve the best prediction performance. Therefore, the motivation of this work is totally unclear.

3. The paper does not provide any theoretical proof to support the findings or the motivations. The relaxation used in the unified framework is also misleading.

**Questions:**

1. Why can we use the relaxation of the binary vector t to its convex hull?
2. How does the proposed method solve the fairness issue from the theoretical aspect?
3. What is the formal definition of the structure bias in graphs, and how can we quantify it?
4. Why do we use active learning to solve the fairness issue in graphs? What if we do not have access to the oracle?

---

> ### Author Response · Authors · 2023-11-18
> **Response to Reviewer tqC2 - Part 1**
>
> Dear Reviewer tqC2,
>
> We appreciate your constructive feedback. We are pleased to provide detailed responses to address your concerns.
>
> > **Q1**: The paper tries to solve the structure fairness problem using active learning. However, the connection between these two is weak, and the reviewer does not find any strong motivations to do so.
>
> **R1**: Thank you for your feedback. We would like to clarify that the main focus of this paper is the active learning for graph neural networks, which aims to maximize the model performance within a labeling budget. There are two main reasons why we connect the structure fairness problem with active learning. **First**, the structure fairness discussed in our paper is based on the labeling. We explore "Label Position Bias" in GNNs, a critical performance fairness issue arising from varying performance based on nodes' "structural distance" to labeled nodes. The distance can be measured by the Label Proximity Score (LPS). Research [1] indicates that nodes closer to labeled nodes (higher LPS) typically perform better. They found the low variance of the LPS score, which means all the nodes have a similar "distance" to the labeled node, can mitigate the label position bias and improve the performance fairness of GNNs. Thus, we use the LPS variance to measure structural fairness. One natural way of reducing the LPS variance is to strategically select the labeled nodes based on their position in the graph.  Meanwhile, the goal of active learning is to select proper nodes for annotation that can maximize the model performance. Thus, **both problems are related to the labeling. Second**, from our preliminary studies, we found that structural fairness (low LPS Variance) can improve the overall model performance. As a result, we use the LPS variance as one metric to help active learning. Meanwhile, as the structural fairness (low LPS Variance) can improve the performance fairness, we find that the proposed method can improve both the model performance and the performance fairness of GNNs.
>
> [1] Towards Label Position Bias in Graph Neural Networks, NeurIPS'23
>
> > **Q2**: What is the formal definition of the structure bias in graphs, and how can we quantify it?
>
> **R2**: **Label Position Bias & Structural Fairness**:  Label Position Bias arises when the performance of a GNN model varies for nodes based on their proximity to labeled nodes in the graph.[1] Formally, in a graph G, with the set of labeled nodes $V_L$, and unlabeled nodes $V_U$, a GNN model trained on $V_L$ may exhibit varying performance for two nodes $i, j \in V_U $ based on their "structural distance" to $V_L$. A key metric we use to measure this distance is the Label Proximal Score $LPS_{i} = \sum_{j \in V_L} P_{ij}$, where P is the personalized PageRank matrix. The LPS can well quantify the proximity of unlabeled nodes to labeled nodes and a higher LPS score typically correlates with better model performance for that node [1].  We use the variance of the LPS score of each node as the measure of structural fairness.
>
> **Measuring Performance Fairness**: To quantify performance fairness, we categorize nodes into different sensitive groups based on their LPS scores. For each group, we calculate the average accuracy of the GNNs and use metrics such as Standard Deviation to measure the performance disparity across these groups.
>
> > **Q3**: The goal of active learning is different from mitigating the bias in graphs and the ultimate goal of AL is to use as few as labeled nodes to achieve the best prediction performance. Therefore, the motivation of this work is totally unclear.
>
> **R3**: The label position bias studied in this paper is different from the traditional attribute bias issue. It is unique for graphs and is related to the labeled nodes' position in the graphs. A natural solution to mitigate the label position bias is to select the nodes, that have proper positions in the graph, to label. Meanwhile, as you mentioned, the ultimate goal of AL is to select the proper nodes to label to achieve the best prediction performance. As a result, both active learning and mitigating the bias are fundamentally related to the labeling of nodes in a graph.
>
> In the preliminary studies, we found that structure fairness (Low LPS variance), which can reflect the label position bias, is also related to the overall performance of GNNs. Therefore, we leverage the LPS variance as one metric in the active learning to improve the overall model performance. This bridges the gap between active learning and mitigating the label position bias in GNNs, proposing that addressing fairness can be synergistic with achieving high model performance.

---

> ### Author Response · Authors · 2023-11-18
> **Response to Reviewer tqC2 - Part 2**
>
> > **Q4**: The paper does not provide any theoretical proof to support the findings or the motivations.
>
> **R4**: We appreciate the concern regarding the lack of theoretical proof in our paper. Our research, focusing on the effectiveness of our approach in the context of active learning for GNNs, is primarily grounded in empirical evaluations. This decision is partly informed by the nature of active learning itself, which, as suggested by the No Free Lunch Theorem, does not subscribe to a one-size-fits-all strategy. Active learning metrics often vary significantly based on the specific datasets and contexts they are applied to. As a result, most active learning approaches use some heuristic metrics to select samples [2][3].
>
> To illustrate this, let's consider the example of the FeatProp method, which is based on the assumption of linearity in GNNs. This method presupposes that the original features can represent the final node representations in GNNs. However, this assumption often falls short due to the inherent non-linearity of neural networks. As such, in scenarios where the original features are compromised – for instance, in the presence of noise or feature missing – the FeatProp method's performance can be significantly impacted, as shown in the experiments in Appendix Section B.3 of our revised paper.  For your convenience, we also present partial results here:
> 1. Noise feature setting: we add different levels of gaussion noise to the original feature.
> |GCN|||Cora|||||CiteSeer|||
> |:---:|:---:|:---:|:---:|:---:|:---:|:---:|:---:|:---:|:---:|:---:|
> |Noise|0.1|0.3|0.5|0.7|0.9|0.1|0.3|0.5|0.7|0.9|
> |Random|64.52±4.50|51.76±4.19|44.76±4.12|40.54±4.12|38.65±3.33|49.16±8.22|36.36±5.71|29.53±4.38|27.05±2.07|26.34±1.85|
> |Degree|61.58±0.44|54.84±0.66|51.25±0.60|49.82±0.62|47.37±0.79|44.37±0.88|37.27±0.47|34.28±0.56|34.53±0.46|34.12±0.56|
> |Pagerank|60.77±1.16|43.22±2.59|34.96±2.23|32.86±0.94|32.76±1.27|49.34±2.77|33.40±2.49|27.99±1.22|25.41±1.95|25.08±1.74|
> |FeatProp|77.51±0.61|45.93±2.18|38.50±1.63|32.94±0.48|26.86±1.45|52.30±2.47|33.16±1.62|27.59±1.84|26.10±1.25|24.88±0.98|
> |GraphPart|79.22±0.48|62.17±1.24|51.80±0.85|45.01±1.05|40.83±1.56|52.37±1.13|44.63±1.05|38.91±0.66|35.59±0.92|34.42±1.51|
> |SCARCE|75.13±0.81|67.23±0.70|59.42±0.47|55.90±0.63|53.45±0.96|59.50±2.53|50.35±0.56|41.61±0.83|37.23±0.63|35.23±1.11|
> 2. Missing feature setting: we use the one-hot ID as node features.
>  |GCN||Cora|||CiteSeer||
> |:---:|:---:|:---:|:---:|:---:|:---:|:---:|
> |Budget|5C|10C|20C|5C|10C|20C|
> |Random|44.76±5.83|56.47±2.30|64.03±1.74|28.41±2.69|34.78±3.06|40.58±2.61|
> |Degree|44.19±7.98|51.69±2.82|63.26±1.95|27.33±1.97|31.74±2.79|40.99±2.02|
> |Pagerank|39.19±3.55|47.95±3.94|63.93±3.85|24.03±1.96|32.83±2.21|40.35±2.02|
> |FeatProp|47.75±3.16|58.84±3.22|65.66±1.38|29.79±2.32|34.79±2.00|41.89±2.71|
> |GraphPart|55.37±4.06|57.40±4.24|67.60±2.61|38.27±2.08|37.40±2.56|43.61±3.15|
> |SCARCE|61.43±2.31|68.64±1.94|69.59±1.22|37.89±1.94|43.51±1.44|48.20±1.15|
>
> While we acknowledge the importance of theoretical backing in scientific research, our current focus has been on demonstrating the practical efficacy and adaptability of our approach in diverse and realistic settings. Moving forward, we aim to complement our empirical findings with theoretical analysis to provide a more holistic understanding of our approach's mechanics and its broader applicability.
>
> [2] A Survey of Deep Active Learning, CSUR'21
>
> [3] A Comparative Survey of Deep Active Learning, Arxiv'22
>
> > **Q5**: The relaxation used in the unified framework is also misleading. Why can we use the relaxation of the binary vector t to its convex hull?
>
> **R5**:  The relaxation of binary vectors to their convex hull is a well-established technique in combinatorial optimization [4], particularly when dealing with complex optimization problems that are otherwise computationally intractable. This approach allows for the application of more efficient, continuous optimization methods. This method is usually used in gradient-based attacks [5][6]. For example, [5] leverages this relaxation to do the topology attack on graphs.
>
> [4] LP Relaxations for Combinatorial Relaxation
>
> [5] Topology attack and defense for graph neural networks: An optimization perspective, Arxiv'19
>
> [6] Probabilistic Categorical Adversarial Attack and Adversarial Training, ICML'23

---

> ### Author Response · Authors · 2023-11-18
> **Response to Reviewer tqC2 - Part 3**
>
> > **Q6**: How does the proposed method solve the fairness issue from the theoretical aspect?
>
> **R6**: In this work, we focus on the performance fairness issue caused by the labeling of nodes, which is different from the traditional fairness issue caused by some sensitive attributes, such as gender and race. To solve this performance fairness issue, we directly optimize the structure fairness (low LPS variance) by selecting the nodes for annotation (active learning). And we empirically demonstrate that the proposed method can mitigate the performance fairness issue, as demonstrated in Figures 4 and 5 of our paper. Besides, the LPS score of node $y$ is related to the influence score from labeled node $x$, defined as the absolute values of entries of Jacobian matrix $\left[\frac{\partial h_x^{(k)}}{\partial h_y^{(0)}}\right]$ [7], where $h_x^{(k)}$ is the representation of node $x$ in a $k$-th layer GNN. [1] found that a lower LPS variance suggests a more uniform influence of all labeled nodes to each unlabeled node across the graph, hinting at greater fairness.
>
> [7] Representation Learning on Graphs with Jumping Knowledge Networks, ICML'18
>
> > **Q7**: Why do we use active learning to solve the fairness issue in graphs? What if we do not have access to the oracle?
>
> **R7**: As mentioned in **R1** and **R3**, both active learning and structure fairness issues are related to the labeling of nodes in a graph. Thus, we leverage the principles of active learning - strategic node selection for labeling - to improve structure fairness and thus solve the performance fairness issue of GNNs.
>
> Regarding the concern about reliance on an oracle for annotations: in active learning, the oracle is typically a necessary component for providing labels for the selected samples. If we do not have access to the oracle, we might need some pre-trained model to do annotation. For example, we can use the Large Language Model as an annotator, like [8].
>
> [8] Exploring the Potential of Large Language Models (LLMs) in Learning on Graphs. Arxiv'23
>
> We hope that we have addressed the concerns in your comments, and please kindly let us know if there is any further concern, and we are happy to clarify.

---

> ### Author Response · Authors · 2023-11-20
> **A Gentle Remind to Reviewer tqC2**
>
> Dear Reviewer tqC2,
>
> We would like to express our sincere gratitude to you for reviewing our paper and providing valuable feedback. Could we kindly know if the responses have addressed your concerns? If there are any further questions, we are happy to clarify. Thank you.
>
> Best,
>
> All authors

---

> > ### Comment · Reviewer_tqC2 · 2023-11-23
> >
> > Thanks for the detailed rebuttal. Most of my concerns have been addressed and I have my score raised.

---

> > > ### Author Response · Authors · 2023-11-23
> > > **Thanks for your response and support**
> > >
> > > Dear Reviewer tqC2,
> > >
> > > Thanks for your response and support. We are glad to know that our rebuttal has addressed your concerns. Please let us know in case there remain outstanding concerns, and if so, we will be happy to respond.
> > >
> > > Best Regards,
> > >
> > > All Authors

---

### Official Review · Reviewer_UJDy · 2023-10-31

**Soundness:** 3 good
**Presentation:** 3 good
**Contribution:** 3 good
**Rating:** 6
**Confidence:** 3

**Summary:**

This paper proposes a unified optimization framework for active learning on graph neural networks (GNNs) that can flexibly incorporate different selection criteria such as structure inertia score (SIS) and label proximity score (LPS) variance. It is empirically demonstrated that SCARCE outperforms existing baselines on node classification tasks across multiple benchmark datasets. In particular, SCARCE achieves higher accuracy than methods like FeatProp and GraphPart while also enhancing fairness by reducing variance in LPS across nodes.

**Strengths:**

(1) This paper is generally well-organized and easy-to-follow.

(2) The proposed unified optimization framework is flexible and does not require extensive hyperparameter tuning, which is especially useful for active learning. In addition, the scalability seems promising as well.

(3) The superiority on utility and fairness seems significant given the presented results in section 4.3 and 4.4.

**Weaknesses:**

(1) There lacks a formal introduction of the notion for fairness at the beginning of this paper.

(2) Despite the discussion on scalability, this paper does not perform any experiments on large-scale network datasets.

(3) Only performing experiments on two GNN backbones undermines the superiority of the proposed framework. In addition, one advantage of this paper lies in the applicability on featureless networks, which is not tested in this paper either.

**Questions:**

(1) I would suggest to add a formal introduction about the fairness notion studied in this paper in Section 2, and add a descriptive discussion in the Introduction accordingly.

(2) If the proposed framework can be easily generalized onto large network data, will the performance superiority still be maintained?

(3) If the proposed framework can be easily generalized onto featureless network data, will the performance superiority still be maintained? Note that in such cases, the feature input of GNNs can be generated following traditional ways.

(4) Can the proposed framework achieve generally good performance across different state-of-the-art GNN backbones? It would be better to adopt more backbones for experiments.

---

> ### Author Response · Authors · 2023-11-18
> **Response to Reviewer UJDy - Part 1**
>
> Dear Reviewer UJDy,
>
> Thank you for your insightful suggestions. We are pleased to provide detailed responses to address your concerns.
>
> > **Q1 & W1**: There lacks a formal introduction of the notion for fairness at the beginning of this paper. (I would suggest to add a formal introduction about the fairness notion studied in this paper in Section 2, and add a descriptive discussion in the Introduction accordingly.)
>
> **R1**: In response to your valuable feedback, we have incorporated a comprehensive introduction of the structure fairness issue in both the Introduction and Section 2 of our revised paper.  For your convenience, we present the formal introduction to the fairness issue discussed in our paper here:
>
> **Label Position Bias & Structural Fairness**: In this paper, we use the notion of  "Label Position Bias", a critical aspect of performance bias in GNNs. This bias arises when the performance of a GNN model varies for nodes based on their proximity to labeled nodes in the graph.[1] Formally, in a graph G, with the set of labeled nodes $V_L$, and unlabeled nodes $V_U$, a GNN model trained on $V_L$ may exhibit varying performance for two nodes $i, j \in V_U $ based on their "structural distance" to $V_L$. A key metric we use to measure this distance is the Label Proximal Score $LPS_{i} = \sum_{j \in V_L} P_{ij}$, where P is the personalized PageRank matrix. The LPS can well quantify the proximity of unlabeled nodes to labeled nodes and a higher LPS score typically correlates with better model performance for that node [1]. [1] also found the low variance of the LPS score, which indicates all the nodes have a similar "distance" to the labeled nodes, can mitigate the label position bias and improve the performance fairness of GNNs. As a result, we use the variance of the LPS score of each node as the measure of structural fairness.
>
> **Measuring Performance Fairness**: To quantify performance fairness, we categorize nodes into different sensitive groups based on their LPS scores. For each group, we calculate the average accuracy of the GNNs and use metrics such as Standard Deviation to measure the performance disparity across these groups.
>
> [1] Towards Label Position Bias in Graph Neural Networks, NeurIPS'23
>
> > **Q2 & W2**: Despite the discussion on scalability, this paper does not perform any experiments on large-scale network datasets.
>
> **R2**: To verify the scalability of the proposed method, in addition to the OGB Arxiv dataset used in our paper, we have conducted extensive experiments on the OGB Products dataset, which contains over 2.4 million nodes. We conduct experiments with the labeling budgets of 5C, 10C, and 20C, where C is the number of classes. The results of the OGB Products dataset with GCN and APPNP backbones are shown below:
>
> |Backbone||GCN|||APPNP||
> |---|---|---|---|---|---|---|
> |Budget|5C|10C|20C|5C|10C|20C|
> |Random|60.78±0.60|67.38±0.74|70.45±0.89|66.15±1.43|70.23±0.42|74.82±0.10|
> |Degree|37.13±0.46|45.82±0.38|50.19±0.99|49.98±0.85|55.43±0.74|60.81±0.97|
> |Pagerank|56.33±0.60|63.20±0.20|69.00±0.45|61.08±0.32|67.61±0.70|72.69±0.18|
> |FeatProp|63.34±2.58|67.48±0.97|71.65±0.57|68.64±0.65|69.80±1.20|71.8±0.61|
> |GraphPart|OOT|OOT|OOT|OOT|OOT|OOT|
> |SCARCE|62.25±0.37|70.27±0.22|74.43±0.22|66.66±0.09|72.63±0.20|76.57±0.14|
>
> Please note the GraphPart method takes more than 24 hours, and we treat it as out of time (OOT).
>
> The results from our experiments on the large-scale OGB Products dataset clearly demonstrate the efficacy of our proposed SCARCE method. Notably, SCARCE achieves high accuracy with significantly fewer labeled nodes compared to standard training splits. For instance, while the standard split typically uses around 200,000 nodes for training, SCARCE requires only 940 labeled nodes to attain accuracies of 74.43/(75.64) and 76.57/(76.62) for the GCN and APPNP models, respectively, where the value in the $(\cdot)$ represent the accuracy of the standard split. This underscores SCARCE's effectiveness in active learning, particularly in large-scale graph neural network applications.

---

> ### Author Response · Authors · 2023-11-18
> **Response to Reviewer UJDy - Part 2**
>
> > **W3 & Q4**: Only performing experiments on two GNN backbones undermines the superiority of the proposed framework.
>
> **R3**: We appreciate your suggestions. We further add two representative GNNs: GAT[2], which leverages the attention mechanism, and GCNII[3], which is a deep GNN. For these two methods, we follow the hyperparameter settings in their original paper. Specifically, for GCNII, we adopt 64 layers. The results on the Cora and CiteSeer datasets with budgets of 5C, 10C and 20 C, are shown below:
>
> |GAT||Cora|||CiteSeer||
> |:---:|:---:|:---:|:---:|:---:|:---:|:---:|
> |Budget|5C|10C|20C|5C|10C|20C|
> |Random|65.66±3.70|74.06±4.19|80.00±2.24|51.36±7.53|61.85±2.34|64.82±2.15|
> |Uncertainty|55.31±5.78|62.02±6.56|74.72±3.69|44.41±6.35|49.95±5.64|60.49±4.00|
> |Density|64.10±4.28|70.72±3.73|78.12±2.97|49.44±7.58|60.10±3.73|64.17±2.02|
> |CoreSet|60.72±7.03|72.54±3.40|79.48±1.52|47.29±6.76|58.00±4.21|64.54±2.11|
> |Degree|62.34±2.20|73.44±1.39|78.31±0.98|42.18±1.89|44.41±1.29|49.38±2.20|
> |Pagerank|63.78±2.05|71.84±1.77|80.00±1.03|42.98±2.40|58.62±1.87|66.51±0.77|
> |AGE|66.45±4.71|73.93±3.89|80.14±1.55|49.53±5.40|58.97±5.94|65.18±2.26|
> |FeatProp|73.40±1.39|77.62±1.36|82.36±0.53|58.03±2.99|62.14±2.29|68.08±0.54|
> |GraphPart|76.91±1.41|78.29±0.88|80.94±1.52|58.24±2.41|65.10±1.32|66.41±1.33|
> |SCARCE-Structure|76.36±2.00|81.78±0.46|82.58±0.41|64.23±1.92|69.32±0.60|72.36±0.49|
> |SCARCE-Feature|80.56±0.77|83.01±0.63|85.04±0.26|60.91±3.10|70.47±0.61|72.60±0.38|
> |SCARCE-ALL|80.62±0.50|83.27±0.39|84.96±0.24|61.20±2.77|70.29±0.61|72.34±0.40|
>
> |GCNII||Cora|||CiteSeer||
> |:---:|:---:|:---:|:---:|:---:|:---:|:---:|
> ||5C|10C|20C|5C|10C|20C|
> |Random|69.76±6.47|78.29±3.99|83.24±1.08|55.25±7.68|65.59±1.60|68.25±1.28|
> |Uncertainty|68.78±5.64|77.15±3.36|82.33±1.30|53.59±8.70|62.91±5.47|67.97±1.69|
> |Density|72.31±3.10|78.85±1.61|83.22±1.08|55.49±7.04|65.41±2.80|68.07±1.51|
> |CoreSet|66.07±3.34|76.11±3.37|80.43±1.97|54.22±6.17|61.45±3.05|67.41±1.02|
> |Degree|68.02±1.92|77.92±0.45|82.85±0.41|58.99±1.61|60.62±1.95|67.41±0.50|
> |Pagerank|69.82±2.74|76.64±1.27|83.52±0.51|53.41±2.70|62.61±1.00|69.53±0.66|
> |AGE|71.31±3.89|78.81±2.14|83.29±1.23|56.65±4.73|64.46±2.58|67.59±2.60|
> |FeatProp|74.63±2.59|83.85±0.58|84.49±0.60|44.99±2.25|52.97±2.08|63.01±0.93|
> |GraphPart|80.45±1.07|82.47±0.46|83.4±0.42|47.89±2.25|58.99±1.34|65.99±1.08|
> |SCARCE-Structure|79.96±1.10|83.43±0.54|83.98±0.48|64.55±0.72|69.99±0.74|72.05±0.35|
> |SCARCE-Feature|82.05±1.32|83.25±0.65|85.96±0.53|67.63±1.98|70.98±0.37|72.25±0.74|
> |SCARCE-ALL|82.61±1.01|83.15±0.47|85.94±0.51|66.37±4.07|70.91±0.42|72.89±0.44|
>
> From the above results, we can observe that the proposed SCARCE can also work well with both GAT and GCNII backbones. This result demostrate the proposed SCARCE can achieve generally good performance across different state-of-the-art GNN backbones.
>
> [2] Graph attention networks. ICLR'18
>
> [3] Simple and deep graph convolutional networks. ICML'20
>
> > **Q3**: In addition, one advantage of this paper lies in the applicability on featureless networks, which is not tested in this paper either.
>
> **R4**: Thanks for your suggestion. To verify the effectiveness of the proposed method on low-quality feature cases, we conducted additional experiments in two distinct settings, i.e., Noise feature and Missing feature setting.
>
> 1. **Noise Feature Setting**: We introduced varying levels of standard Gaussian noise to the original features. Specifically,  we modified the features as $X = X + k*\epsilon$, where $\epsilon \sim \mathcal{N}(0,1)$, and k takes values from the $[0.1, 0.3, 0.5, 0.7, 0.9]$. This experiment was conducted on the Cora and CiteSeer datasets with budget settings of 5C and 10C, where C represents the number of classes in the dataset. The results are shown in Appendix Section B.3 in our revised paper. For your convenience, we present the results of 5C budgets with GCN and APPNP backbones in Part 3.

---

> ### Author Response · Authors · 2023-11-18
> **Response to Reviewer UJDy - Part 3**
>
> **R4 Continued**: The results of the noise feature setting are shown below:
>
> |GCN|||Cora|||||CiteSeer|||
> |:---:|:---:|:---:|:---:|:---:|:---:|:---:|:---:|:---:|:---:|:---:|
> |Noise|0.1|0.3|0.5|0.7|0.9|0.1|0.3|0.5|0.7|0.9|
> |Random|64.52±4.50|51.76±4.19|44.76±4.12|40.54±4.12|38.65±3.33|49.16±8.22|36.36±5.71|29.53±4.38|27.05±2.07|26.34±1.85|
> |Degree|61.58±0.44|54.84±0.66|51.25±0.60|49.82±0.62|47.37±0.79|44.37±0.88|37.27±0.47|34.28±0.56|34.53±0.46|34.12±0.56|
> |Pagerank|60.77±1.16|43.22±2.59|34.96±2.23|32.86±0.94|32.76±1.27|49.34±2.77|33.40±2.49|27.99±1.22|25.41±1.95|25.08±1.74|
> |FeatProp|77.51±0.61|45.93±2.18|38.50±1.63|32.94±0.48|26.86±1.45|52.30±2.47|33.16±1.62|27.59±1.84|26.10±1.25|24.88±0.98|
> |GraphPart|79.22±0.48|62.17±1.24|51.80±0.85|45.01±1.05|40.83±1.56|52.37±1.13|44.63±1.05|38.91±0.66|35.59±0.92|34.42±1.51|
> |SCARCE|75.13±0.81|67.23±0.70|59.42±0.47|55.90±0.63|53.45±0.96|59.50±2.53|50.35±0.56|41.61±0.83|37.23±0.63|35.23±1.11|
>
> |APPNP|||Cora|||||CiteSeer|||
> |:---:|:---:|:---:|:---:|:---:|:---:|:---:|:---:|:---:|:---:|:---:|
> |Noise|0.1|0.3|0.5|0.7|0.9|0.1|0.3|0.5|0.7|0.9|
> |Random|69.52±5.14|60.06±4.43|55.21±4.19|53.16±3.60|51.51±4.42|50.91±7.05|39.35±4.37|35.08±5.96|32.97±3.49|31.77±4.15|
> |Degree|67.59±0.92|61.86±0.55|58.98±0.67|56.29±0.82|55.81±0.88|50.83±1.58|41.98±1.10|39.77±0.62|36.92±0.38|36.84±0.51|
> |Pagerank|70.87±1.70|54.90±2.85|50.41±1.87|44.75±1.93|43.62±2.49|52.08±2.55|41.20±2.01|34.63±2.43|30.68±2.27|31.78±1.99|
> |FeatProp|80.17±0.77|48.23±1.72|44.64±1.36|35.49±0.77|35.76±0.86|54.49±1.66|36.12±1.34|27.69±1.57|25.43±0.81|24.31±1.53|
> |GraphPart|80.68±0.41|68.76±1.80|66.11±1.05|59.14±2.04|54.07±2.22|60.38±1.40|45.39±1.36|40.97±1.14|39.19±1.78|38.79±1.11|
> |SCARCE|79.39±0.67|75.89±0.66|72.46±0.20|70.10±1.21|69.19±0.78|62.58±1.24|54.76±0.82|48.60±0.63|45.67±0.55|44.28±0.60|
>
> From the above results, we can have the following observations:
> * With the level of noise increase in the features, the performance of the feature-based methods drops a lot. For example, the FeatProp performs even worse than random selection.
> * The proposed SCARCE, which only leverages the graph structure in active learning, can outperform baselines by a large margin.
>
> 2. **Missing Feature Setting**: To further test the resilience of SCARCE under challenging feature conditions, we conducted experiments in a setting where all features are missing. In this scenario, we replaced the original node features with one-hot ID features. We conduct experiments on both Cora and CiteSeer datasets with labeling budgets of 5C, 10C, and 20C using both GCN and APPNP models. The results are shown below:
>
> |GCN||Cora|||CiteSeer||
> |:---:|:---:|:---:|:---:|:---:|:---:|:---:|
> |Budget|5C|10C|20C|5C|10C|20C|
> |Random|44.76±5.83|56.47±2.30|64.03±1.74|28.41±2.69|34.78±3.06|40.58±2.61|
> |Degree|44.19±7.98|51.69±2.82|63.26±1.95|27.33±1.97|31.74±2.79|40.99±2.02|
> |Pagerank|39.19±3.55|47.95±3.94|63.93±3.85|24.03±1.96|32.83±2.21|40.35±2.02|
> |FeatProp|47.75±3.16|58.84±3.22|65.66±1.38|29.79±2.32|34.79±2.00|41.89±2.71|
> |GraphPart|55.37±4.06|57.40±4.24|67.60±2.61|38.27±2.08|37.40±2.56|43.61±3.15|
> |SCARCE|61.43±2.31|68.64±1.94|69.59±1.22|37.89±1.94|43.51±1.44|48.20±1.15|
>
> |APPNP||Cora|||CiteSeer||
> |:---:|:---:|:---:|:---:|:---:|:---:|:---:|
> |Budget|5C|10C|20C|5C|10C|20C|
> |Random|62.27±3.86|68.71±3.22|75.52±1.65|38.24±3.21|46.10±2.85|49.98±1.52|
> |Degree|64.48±2.97|65.29±1.63|73.21±1.17|38.92±4.79|46.93±1.60|49.10±2.16|
> |Pagerank|61.79±1.61|68.89±1.11|74.15±0.58|31.94±1.20|42.58±1.39|50.70±1.05|
> |FeatProp|35.22±0.99|47.42±3.34|72.47±0.42|31.89±0.46|31.98±0.57|29.01±0.56|
> |GraphPart|66.40±0.60|71.98±0.59|74.15±1.38|46.10±0.62|49.93±0.40|51.34±0.78|
> |SCARCE|72.39±1.54|76.33±0.99|76.77±0.71|48.06±3.66|52.20±2.02|56.21±1.92|
>
> From the above results, we can have similar findings: While the feature-based methods, such as FeatProp, perform poorly under the missing feature setting, our proposed SCARCE can achieve good performance.
>
> In summary, the proposed method SCARCE can perform as expected when the feature quality is low or missing, but the feature-based active learning methods may perform poorly.
>
> We hope that we have addressed the concerns in your comments, and please kindly let us know if there is any further concern, and we are happy to clarify.

---

> ### Author Response · Authors · 2023-11-20
> **A Gentle Remind to Reviewer UJDy**
>
> Dear Reviewer UJDy,
>
> We would like to express our sincere gratitude to you for reviewing our paper and providing valuable feedback. Could we kindly know if the responses have addressed your concerns? If there are any further questions, we are happy to clarify. Thank you.
>
> Best,
>
> All authors

---

> ### Comment · Reviewer_UJDy · 2023-11-23
> **Thanks for the rebuttal**
>
> Thanks for the detailed rebuttal. Most of my concerns have been addressed and I have my score raised.

---

> > ### Author Response · Authors · 2023-11-23
> > **Thanks for your response and support**
> >
> > Dear Reviewer UJDy,
> >
> > Thanks for your response and support. We are glad to know that our rebuttal has addressed your concerns. Please let us know in case there remain outstanding concerns, and if so, we will be happy to respond.
> >
> > Best Regards,
> >
> > All Authors

---

### Official Review · Reviewer_BDeb · 2023-11-06

**Soundness:** 2 fair
**Presentation:** 3 good
**Contribution:** 3 good
**Rating:** 6
**Confidence:** 4

**Summary:**

Existing active learning models for GNNs heavily rely on the quality of initial node features and ignore the impact of label position bias in the selection of representative nodes. To address these limitations, this paper proposes a novel framework called SCARCE.

**Strengths:**

+ They identify the limitations in current active learning methods, specifically the oversight regarding feature quality and position bias.
+ They propose a novel framework to tackle the aforementioned limitations.
+ Extensive experiments validate the effectiveness of the proposed framework.

**Weaknesses:**

- There are concerns regarding the fundamental motivation behind active learning. While the primary motivation for active learning lies in the difficulty of obtaining high-quality labels in real-world scenarios, the iterative addition of labels for learned target nodes during the optimization process raises doubts about the original motivation. This creates some contradiction as it suggests that labels for target nodes might be easy to obtain.
- The improvement compared to baselines seems not statistically significant.
- They argue that existing methods heavily rely on the quality of initial node features while the proposed framework can mitigate this problem. However, there lack of experimental support. The features of datasets seem typical, lacking any characteristics such as unavailability and noise. There is a need for a quantifiable evaluation to support this point.

**Questions:**

Please refer to the weaknesses.
- There are concerns regarding the fundamental motivation behind active learning. While the primary motivation for active learning lies in the difficulty of obtaining high-quality labels in real-world scenarios, the iterative addition of labels for learned target nodes during the optimization process raises doubts about the original motivation. This creates some contradiction as it suggests that labels for target nodes might be easy to obtain.
- The improvement compared to baselines seems not statistically significant.
- They argue that existing methods heavily rely on the quality of initial node features while the proposed framework can mitigate this problem. However, there lack of experimental support. The features of datasets seem typical, lacking any characteristics such as unavailability and noise. There is a need for a quantifiable evaluation to support this point.

---

> ### Author Response · Authors · 2023-11-18
> **Response to reviewer BDeb - Part 1**
>
> Dear Reviewer BDeb,
>
> We appreciate your constructive feedback. We are pleased to provide detailed responses to address your concerns.
>
> > **Q1**: There are concerns regarding the fundamental motivation behind active learning. While the primary motivation for active learning lies in the difficulty of obtaining high-quality labels in real-world scenarios, the iterative addition of labels for learned target nodes during the optimization process raises doubts about the original motivation. This creates some contradiction as it suggests that labels for target nodes might be easy to obtain.
>
> **R1**:  Thank you for pointing out this concern. We would like to clarify that the fundamental motivation behind active learning is that we want to maximize the model's performance within a limited labeling budget. Due to the difficulty of obtaining high-quality labels, we usually assume the samples are labeled by human experts (often called “oracle”), which can be expensive and time-consuming. Therefore, we usually set a labeling budget to constrain the number of samples that can be labeled by human experts.
>
> In active learning, there are primarily two settings: iterative [1][2] and one-step [3][4]. In the iterative setting, as you mentioned, we ask human experts to label a small number of samples and then train a model based on these samples in each iteration. Afterward, we use the learned model to select some new samples (for example, samples with low confidence based on the model) and ask human experts to label them. This process is repeated until the limited budget is reached. Conversely, in the one-step active learning setting, which is the focus of our work, all the labels are obtained at once within the set budget. This approach avoids the resource-intensive process of training the model multiple times. However, for both settings, the effects of human experts are the same as the number of labeled samples is the budget. Therefore, it does not imply that obtaining labels is easy.
>
> [1] An analysis of active learning strategies for sequence labeling tasks.
>
> [2] Active learning for graph embedding.
>
> [3] Active learning for graph neural networks via node feature propagation.
>
> [4] Partition-based active learning for graph neural networks.
>
> > **Q2**: The improvement compared to baselines seems not statistically significant.
>
> **R2**: We have conducted the t-test between our proposed SCARCE and the best baseline model across each dataset and budget setting. The p-value less than 0.05 usually indicates that the improvement is statistically significant. In our paper, we have marked results that are statistically significant with a red asterisk (*) in Tables 1 and 2 for clear reference.  For your convenience, we also present partial results of the GCN model here:
>
> | DataSet |  | Cora |  |  | CiteSeer |  |  | Computer |  |  | Arxiv |  |
> |:---:|:---:|:---:|:---:|:---:|:---:|:---:|:---:|:---:|:---:|:---:|:---:|:---:|
> | Budget | 5C | 10C | 20C | 5C | 10C | 20C | 5C | 10C | 20C | 5C | 10C | 20C |
> | P-Value | 2.9E-06 | 0.4 | 1.1E-04 | 8.0E-10 | 9.7E-19 | 4.4E-20 | 3.0E-06 | 0.01 | 4.7E-05 | 1.2E-06 | 1.5E-10 | 2.3E-21 |
>
> From the results, the majority of our improvements over baseline models are indeed statistically significant.
>
> > **Q3**: They argue that existing methods heavily rely on the quality of initial node features while the proposed framework can mitigate this problem. However, there lack of experimental support. The features of datasets seem typical, lacking any characteristics such as unavailability and noise. There is a need for a quantifiable evaluation to support this point.
>
> **R3**: Thank you for this great suggestion. Following your suggestion, we conducted additional experiments in two distinct settings to demonstrate the superiority of the proposed SCARCE.
>
> 1. **Noise Feature Setting**: We introduced varying levels of standard Gaussian noise to the original features. Specifically, we modified the features as $X = X + k*\epsilon$, where $\epsilon \sim \mathcal{N}(0,1)$, and k takes values from the $[0.1, 0.3, 0.5, 0.7, 0.9]$. This experiment was conducted on the Cora and CiteSeer datasets with budget settings of 5C and 10C, where C represents the number of classes in the dataset. The results are shown in Appendix B.3 of our revised paper. For your convenience, we present the partial results of 5C budgets with GCN and APPNP backbones in Part 2.

---

> ### Author Response · Authors · 2023-11-18
> **Response to reviewer BDeb - Part 2**
>
> **R3 Continued**: The results of the noise feature setting are shown below:
>
> |GCN|||Cora|||||CiteSeer|||
> |:---:|:---:|:---:|:---:|:---:|:---:|:---:|:---:|:---:|:---:|:---:|
> |Noise|0.1|0.3|0.5|0.7|0.9|0.1|0.3|0.5|0.7|0.9|
> |Random|64.52±4.50|51.76±4.19|44.76±4.12|40.54±4.12|38.65±3.33|49.16±8.22|36.36±5.71|29.53±4.38|27.05±2.07|26.34±1.85|
> |Degree|61.58±0.44|54.84±0.66|51.25±0.60|49.82±0.62|47.37±0.79|44.37±0.88|37.27±0.47|34.28±0.56|34.53±0.46|34.12±0.56|
> |Pagerank|60.77±1.16|43.22±2.59|34.96±2.23|32.86±0.94|32.76±1.27|49.34±2.77|33.40±2.49|27.99±1.22|25.41±1.95|25.08±1.74|
> |FeatProp|77.51±0.61|45.93±2.18|38.50±1.63|32.94±0.48|26.86±1.45|52.30±2.47|33.16±1.62|27.59±1.84|26.10±1.25|24.88±0.98|
> |GraphPart|79.22±0.48|62.17±1.24|51.80±0.85|45.01±1.05|40.83±1.56|52.37±1.13|44.63±1.05|38.91±0.66|35.59±0.92|34.42±1.51|
> |SCARCE|75.13±0.81|67.23±0.70|59.42±0.47|55.90±0.63|53.45±0.96|59.50±2.53|50.35±0.56|41.61±0.83|37.23±0.63|35.23±1.11|
>
> |APPNP|||Cora|||||CiteSeer|||
> |:---:|:---:|:---:|:---:|:---:|:---:|:---:|:---:|:---:|:---:|:---:|
> |Noise|0.1|0.3|0.5|0.7|0.9|0.1|0.3|0.5|0.7|0.9|
> |Random|69.52±5.14|60.06±4.43|55.21±4.19|53.16±3.60|51.51±4.42|50.91±7.05|39.35±4.37|35.08±5.96|32.97±3.49|31.77±4.15|
> |Degree|67.59±0.92|61.86±0.55|58.98±0.67|56.29±0.82|55.81±0.88|50.83±1.58|41.98±1.10|39.77±0.62|36.92±0.38|36.84±0.51|
> |Pagerank|70.87±1.70|54.90±2.85|50.41±1.87|44.75±1.93|43.62±2.49|52.08±2.55|41.20±2.01|34.63±2.43|30.68±2.27|31.78±1.99|
> |FeatProp|80.17±0.77|48.23±1.72|44.64±1.36|35.49±0.77|35.76±0.86|54.49±1.66|36.12±1.34|27.69±1.57|25.43±0.81|24.31±1.53|
> |GraphPart|80.68±0.41|68.76±1.80|66.11±1.05|59.14±2.04|54.07±2.22|60.38±1.40|45.39±1.36|40.97±1.14|39.19±1.78|38.79±1.11|
> |SCARCE|79.39±0.67|75.89±0.66|72.46±0.20|70.10±1.21|69.19±0.78|62.58±1.24|54.76±0.82|48.60±0.63|45.67±0.55|44.28±0.60|
>
> From the above results, we can have the following observations:
> * With the level of noise increase in the features, the performance of the feature-based methods drops a lot. For example, the FeatProp performs even worse than random selection.
> * The proposed SCARCE, which only leverages the graph structure in active learning, can outperform baselines by a large margin.
>
> 2. **Missing Feature Setting**: To further test the resilience of SCARCE under challenging feature conditions, we conducted experiments in a setting where all features are missing. In this scenario, we replaced the original node features with one-hot ID features. We conduct experiments on both Cora and CiteSeer datasets with labeling budgets of 5C, 10C, and 20C using both GCN and APPNP models. The results are shown below:
>
> |GCN||Cora|||CiteSeer||
> |:---:|:---:|:---:|:---:|:---:|:---:|:---:|
> |Budget|5C|10C|20C|5C|10C|20C|
> |Random|44.76±5.83|56.47±2.30|64.03±1.74|28.41±2.69|34.78±3.06|40.58±2.61|
> |Degree|44.19±7.98|51.69±2.82|63.26±1.95|27.33±1.97|31.74±2.79|40.99±2.02|
> |Pagerank|39.19±3.55|47.95±3.94|63.93±3.85|24.03±1.96|32.83±2.21|40.35±2.02|
> |FeatProp|47.75±3.16|58.84±3.22|65.66±1.38|29.79±2.32|34.79±2.00|41.89±2.71|
> |GraphPart|55.37±4.06|57.40±4.24|67.60±2.61|38.27±2.08|37.40±2.56|43.61±3.15|
> |SCARCE|61.43±2.31|68.64±1.94|69.59±1.22|37.89±1.94|43.51±1.44|48.20±1.15|
>
> |APPNP||Cora|||CiteSeer||
> |:---:|:---:|:---:|:---:|:---:|:---:|:---:|
> |Budget|5C|10C|20C|5C|10C|20C|
> |Random|62.27±3.86|68.71±3.22|75.52±1.65|38.24±3.21|46.10±2.85|49.98±1.52|
> |Degree|64.48±2.97|65.29±1.63|73.21±1.17|38.92±4.79|46.93±1.60|49.10±2.16|
> |Pagerank|61.79±1.61|68.89±1.11|74.15±0.58|31.94±1.20|42.58±1.39|50.70±1.05|
> |FeatProp|35.22±0.99|47.42±3.34|72.47±0.42|31.89±0.46|31.98±0.57|29.01±0.56|
> |GraphPart|66.40±0.60|71.98±0.59|74.15±1.38|46.10±0.62|49.93±0.40|51.34±0.78|
> |SCARCE|72.39±1.54|76.33±0.99|76.77±0.71|48.06±3.66|52.20±2.02|56.21±1.92|
>
> From the above results, we can have similar findings: While the feature-based methods, such as FeatProp, perform poorly under the missing feature setting, our proposed SCARCE can achieve good performance.
>
> In summary, the proposed method SCARCE can perform as expected when the feature quality is low or missing, but the feature-based active learning methods may perform poorly.
>
> We hope that we have addressed the concerns in your comments, and please kindly let us know if there is any further concern, and we are happy to clarify.

---

> ### Author Response · Authors · 2023-11-20
> **A Gentle Remind to Reviewer BDeb**
>
> Dear Reviewer BDeb,
>
> We would like to express our sincere gratitude to you for reviewing our paper and providing valuable feedback. Could we kindly know if the responses have addressed your concerns? If there are any further questions, we are happy to clarify. Thank you.
>
> Best,
>
> All authors

---

### Author Response · Authors · 2023-11-22
**A friendly reminder**

Dear Reviewers,

We wish to express our gratitude for your thoughtful comments and concerns. Following your valuable feedback, we have submitted our responses and revisions to address the issues. As the discussion period is about to end, we kindly request your confirmation of the receipt of our responses. Additionally, we welcome any further concerns or suggestions regarding our response. Your timely response is greatly appreciated and will be immensely helpful for us to improve our work.

Thank you for your time and consideration.

Sincerely,

The Authors

---

### Meta-Review · Area_Chair_44Cb · 2023-12-06

**Metareview:**

The submission proposes an active learning algorithm in the graph-based setting, where access to node labels are limited, with the goal of training a graph neural network. The authors argue that standard active learning methods do not consider, and may suffer from, label position bias (the observation that a model can achieve better prediction performance on nodes that are closer to already labeled nodes).

The proposed algorithm is designed with specific attention to this graph structural bias and is shown to empirically outperform several existing active learning methods (general and GNN specific). As a side effect, the authors also find that the proposed algorithm provides improved fairness (in terms of group performance across groups).

Several clarifications, new experiments, and changes to the presentation were made in response to concerns raised by reviewers, which I believe were almost entirely addressed.

**Justification For Why Not Higher Score:**

Based on the reviewer feedback and my own reading, the contributions do not appear to be so groundbreaking as to warrant a spotlight or oral.

**Justification For Why Not Lower Score:**

After clarifications and revision, the submission addresses most of the reviewer concerns and provides an effective algorithm for a setting of interest.

---

### Decision · Program_Chairs · 2024-01-16

Accept (poster)